# *Olea europaea* L. Root Endophyte *Bacillus velezensis* OEE1 Counteracts Oomycete and Fungal Harmful Pathogens and Harbours a Large Repertoire of Secreted and Volatile Metabolites and Beneficial Functional Genes

**DOI:** 10.3390/microorganisms7090314

**Published:** 2019-09-03

**Authors:** Manel Cheffi, Ali Chenari Bouket, Faizah N. Alenezi, Lenka Luptakova, Marta Belka, Armelle Vallat, Mostafa E. Rateb, Slim Tounsi, Mohamed Ali Triki, Lassaad Belbahri

**Affiliations:** 1Institut de l’Olivier Sfax, Sfax 3000, Tunisia (M.C.) (M.A.T.); 2Plant Protection Research Department, East Azarbaijan Agricultural and Natural Resources Research and Education Center, AREEO, Tabriz 5355179854, Iran; 3NextBiotech, 98 Rue Ali Belhouane, 3030 Agareb, Tunisia (F.N.A.) (L.L.); 4Department of Biology and Genetics, Institute of Biology, Zoology and Radiobiology, University of Veterinary Medicine and Pharmacy in Košice, 04181 Košice, Slovakia; 5Department of Forest Pathology, Poznań University of Life Sciences, Wojska Polskiego 71c, 60-628 Poznań, Poland; 6Department of Plant and Soil Science, Institute of Biological and Environmental Sciences, University of Aberdeen, Cruickshank Building, Aberdeen AB24 3UU, UK; 7Neuchâtel Platform of Analytical Chemistry, Institute of Chemistry, University of Neuchatel, 2000 Neuchatel, Switzerland; 8School of Computing, Engineering & Physical Sciences, University of the West of Scotland, Paisley PA1 2BE, UK; 9Laboratory of Biopesticides, Centre of Biotechnology of Sfax, Sfax 3000, Tunisia; 10Laboratory of Soil Biodiversity, University of Neuchatel, 2000 Neuchatel, Switzerland

**Keywords:** endophytes, secondary metabolites, *Bacillus velezensis*, antimicrobial activity, plant growth promoting bacteria, *Fusarium solani*, abiotic stress

## Abstract

Oomycete and fungal pathogens, mainly *Phytophthora* and *Fusarium* species, are notorious causal agents of huge economic losses and environmental damages. For instance, *Phytophthora ramorum*, *Phytophthora cryptogea*, *Phytophthora plurivora* and *Fusarium solani* cause significant losses in nurseries and in forest ecosystems. Chemical treatments, while harmful to the environment and human health, have been proved to have little or no impact on these species. Recently, biocontrol bacterial species were used to cope with these pathogens and have shown promising prospects towards sustainable and eco-friendly agricultural practices. Olive trees prone to *Phytophthora* and *Fusarium* disease outbreaks are suitable for habitat-adapted symbiotic strategies, to recover oomycetes and fungal pathogen biocontrol agents. Using this strategy, we showed that olive trees-associated microbiome represents a valuable source for microorganisms, promoting plant growth and healthy benefits in addition to being biocontrol agents against oomycete and fungal diseases. Isolation, characterization and screening of root microbiome of olive trees against numerous *Phytophthora* and other fungal pathogens have led to the identification of the *Bacillus velezensis* OEE1, with plant growth promotion (PGP) abilities and strong activity against major oomycete and fungal pathogens. Phylogenomic analysis of the strain OEE1 showed that *B. velezensis* suffers taxonomic imprecision that blurs species delimitation, impacting their biofertilizers’ practical use. Genome mining of several *B. velezensis* strains available in the GenBank have highlighted a wide array of plant growth promoting rhizobacteria (PGPR) features, metals and antibiotics resistance and the degradation ability of phytotoxic aromatic compounds. Strain OEE1 harbours a large repertoire of secreted and volatile secondary metabolites. Rarefaction analysis of secondary metabolites richness in the *B. velezenis* genomes, unambiguously documented new secondary metabolites from ongoing genome sequencing efforts that warrants more efforts in order to assess the huge diversity in the species. Comparative genomics indicated that *B. velezensis* harbours a core genome endowed with PGP features and accessory genome encoding diverse secondary metabolites. Gas Chromatography-Mass Spectrometry (GC-MS) analysis of OEE1 Volatile Organic Compounds (VOCs) and Liquid Chromatography High Resolution Mass Spectrometry (LC-HRMS) analysis of secondary metabolites identified numerous molecules with PGP abilities that are known to interfere with pathogen development. Moreover, *B. velezensis* OEE1 proved effective in protecting olive trees against *F. solani* in greenhouse experiments and are able to inhabit olive tree roots. Our strategy provides an effective means for isolation of biocontrol agents against recalcitrant pathogens. Their genomic analysis provides necessary clues towards their efficient implementation as biofertilizers.

## 1. Introduction

Oomycetes are distinct phylogenetic lineage of fungus-like eukaryotic microorganisms that account for notorious pathogens that affect vertebrate animals, fish, insects, crustaceans, plants, etc. [1,2,3,4,5,6]. The genus *Phytophthora* is a plant-damaging oomycete that is responsible for average annual losses and control costs that exceed several billion dollars [7,8]. *Phytophthora* spp. usually attack agricultural and forest ecosystems; their spread is driven by human activities and is exacerbated by climate changes [9,10]. Examples of ecological disasters driven by the *Phytophthora* species include sudden oak death in California [11] and the Mediterranean *Quercus* species thriving in Portugal and South-West Spain [12]. From a Bordeaux mixture to Metalaxyl (Ridomil), the most widely used fungicides against *Phytophthora* spp., several drawbacks appeared in the chemical control of fungi, including the spread of such synthetic fungicides in soil and water ecosystems, in addition to the resistance developed among the *Phytophthora* populations [13,14,15]. Therefore, it could be concluded that effective control of the *Phytophthora* disease would rarely be achieved through fungicide application [13,16]. On the other hand, *Fusarium* diseases have a high incidence on olive trees in the Mediterranean and their chemical control is quite challenging [17]. Emerging biological control approaches provide an attractive, environmentally sound and effective option of reducing or mitigating *Phytophthora* diseases [13,16,18]. This is generally achieved through the recruitment of natural microbial enemies [19].

Habitat-adapted symbiosis driven by plant growth promoting rhizobacteria (PGPR), and more specifically endophytes, which confers pathogen resistance to plants under biotic and abiotic stresses, is a noteworthy option. Several scientists have used a similar strategy and proved that the endophytes of date palm submitted to brittle leaf disease represented an interesting niche of endophytes that are able to control a broad range of phytopathogens [20,21,22]. Additionally, PGPR were shown to confer beneficial effects to plants and considerable tolerance against abiotic stresses [19]. Current means for recovering endophytes are complemented more by phylogenomic analysis and genome mining approaches that provide necessary information towards their efficient implementation as biofertilizers or biocontrol agents [19]. Moreover, these tools are illuminating the basis for strain activity among species [13] and document the large battery of secondary metabolites harboured by their dynamic accessory genome. Such findings warrant more genome sequencing investigations of such strains in order to highlight the diversity of the available secondary metabolites.

In this study, microbiome of olive trees under *Phytophthora* and other fungal infection threats was used as a starting point for antifungal screening of root endophytes against a large collection of *Phytophthora* and other fungal pathogens. The results proved to be effective in the isolation of a bacterial strain that possesses potent inhibitory effects against oomycetes and the fungal pathogens examined and designated as OEE1. Strain OEE1 was able to produce a wide array of VOCs and bioactive secondary metabolites, as suggested by GC-MS and LC-HRMS analyses, respectively. Genome mining of the strain OEE1 showed a core genome with a large repertoire of genes contributing to the plant’s beneficial functions and an accessory genome that harboured a wide secondary metabolite arsenal. Additionally, OEE1 was significantly efficient in the protection of olive trees against *Fusarium solani* infection, in greenhouse experiments.

## 2. Materials and Methods

### 2.1. Bacterial, Oomycete and Fungal Strains and Growth Conditions

The *Bacillus velezensis* strain OEE1 as well as all endophytes used in the study were grown on tryptone soya agar ((TSA); Oxoid AG, Basel, Switzerland). Cell suspensions of the strains were produced by growing them in the tryptone soya broth ((TSB); Oxoid AG, Basel, Switzerland) and incubation at 37 °C for 16 h, with shaking at 180 rpm. OEE1 double resistant strain to streptomycin and rifampicin was generated according to Zouari et al. [23].

The oomycete species was routinely maintained in a V8 medium, as described by Belbahri et al. [1]. Fungi have been maintained on potato dextrose agar plates ((PDA); Oxoid, Basel, Switzerland), according to Khlifi et al. [24]. For preparation of *F. solani* Fso1 spores, fresh culture mycelia of the *F. solani* strain Fso1 was scratched and suspended in a potato dextrose broth (PDB) medium. After growth at 25 °C for 7 days and 100 rpm, conidia concentration was adjusted to 10^6^ conidia mL^−^^1^, using the Malassez Hemocytometer slide and was used in subsequent experiments.

### 2.2. Plant Material

Artificial inoculation and treatments were conducted on 2-years-old olive trees cv. Chemlali, provided by the National Oil Office (ONH, Sfax, Tunisia).

### 2.3. Isolation of Endophytic Bacteria and Screening for Antifungal Activity

Isolation of bacterial endophytes was performed from healthy olive trees belonging to the cultivar Chemlali located in Sfax (South Tunisia, 34°43’50.7” N 10°44’08.6” E). The entire plant was washed under running water to remove soil and dust and endophytic bacteria were isolated from each plant part, as described by [25]. To ensure the surface sterilization efficacy, 100 µL of rinsing water from the last washing was plated on a TSA medium and was incubated at 30 °C [26].

Antifungal activity of the isolated bacteria was evaluated using the dual culture test [17] against *Rhizoctonia bataticola* HQ392809.1, *R. solani* KU863546, *Neofusicoccum australe* EU375516.1, *Nigrospora* sp. JN 207298.1, *Botryospheria* sp., *F. solani* FJ874633.1, Pu: *Pythium ultimum*, *Fusarium oxysporum* JN400698.1 and *Cylindrocarpon* sp. The isolate showing an ability to inhibit the most of pathogens and a strong activity against *F. solani* was selected for the rest of this study.

### 2.4. Antifungal Activity of B. velezensis OEE1 against Oomycetes and True Fungi

*Phytophthora plurivora* MB111501, *P. ramorum* MB121502, *P. citrophthora* MB111304, *P. cactorum* MB120209, *P. cryptogea* MB140718, *P. rosacearum* MB171529 *P. megasperma* MB130527, *Pythium sylvaticum* MB190708, *P. ultimum* MB190423 and *Phytopythium vexans* MB210782, as well as *Fusarium oxysporum* KFL43, *F. sulphureum* MB174817, *F. avenaceum* MB187243, *Cladosporium cladosporioides* MB270438, *F. redolens* MB210208, *C. destructans* KFL7, *Heterobasidion annosum* KFL15, *C. gloeosporioides* MB170211, *Inonotus hispidus* MB180003, *Resinicium bicolor* S150211, *Phaeolus schweinitzii* MB170015, *F. cerealis* MB170015, *Botrytis cinerea* MB170001 and DSM 4709, *F. solani* Fso1 and *Rhizoctonia solani* MB110208 were used in dual culture assays against *B. velezensis* OEE1, to examine its antifungal activity. Bacterial suspension (10 µL) was streaked linearly on one side of a plate of potato dextrose agar ((PDA); Oxoid, Basel, Switzerland) and incubated for 3 days at 37 °C. From an actively growing margin of oomycete and fungal pathogen cultures, agar disks were cut and placed at an equal distance from the periphery of the other side of the petri dish (Figure 1). After incubation at 25°C for 10 days in the dark, the zone between bacterial line and mycelia, which qualified as inhibition zone, was evaluated (expressed in mm) at 24 days intervals [14].

Inhibition of the pathogen growth was estimated following the formula described by Slama et al. [17]:PI = ((D−d)/D) × 100(1)

PI = inhibition of pathogen growth (%); D = diameter of pathogen growth in control plates (mm); d = diameter of pathogen growth for the tests (mm).

The morphological changes of *F. solani* mycelia and conidia were checked by light microscope observations (Leitz WETZLAR, Germany) from the confrontation lines between the OEE1 bacterium and the *F. solani* Fso1 fungus and was compared with the control plate.

### 2.5. Inhibition of F. solani Fso1 Mycelial Growth by Extracellular B. velezensis OEE1 Metabolites

The OEE1 cell-free culture filtrate was used to assess the influence of the extracellular metabolites on *F. solani* radial growth, as described by Slama et al. [17]. The bacterium was grown on an LB broth medium using an orbital shaker (100 rpm), for 72 h. The recovered culture was centrifuged at 4 °C for 5 min at 10,000 rpm. The supernatant was collected and filtered through 0.22 µm membrane filters. The cell-free culture filtrate was added to a warm PDA medium (55 °C) to the final concentrations of 25%, 50% and 75%. PDA plates without the culture filtrates were used as controls. Fungal mycelial plugs of 5 mm diameter were placed centrally in the plates and incubated at 25 °C, until the negative control growth covered the whole surface of the plate. Growth inhibition of the pathogen was measured using the same formula described previously. To determine the fungistatic or fungicidal character of OEE1′s extracellular metabolites against *F. solani*, fungal agar discs exhibiting no growth were selected after the experiment and transferred to new PDA plates without a cell-free culture filtrate. Absence of mycelial growth indicated that the extracellular metabolites had a fungicidal character. While, the mycelial growth indicated the fungistatic character of the OEE1 bacterial extract.

### 2.6. Effect of Extracellular B. velezensis OEE1 Metabolites on F. solani Fso1 Conidia Germination

Different volumes of the culture filtrate were diluted to a final volume of 150 µL, using a soft PDA coloured with a few drops of sterile lactophenol cotton blue. Each concentration (25%, 50% and 75%) was poured as a thin film on the glass slides. Subsequently, 10 µL of the conidia suspension previously adjusted to 10^4^ conidia mL^−1^ were added on the slide surface, after semi-solidification. Negative controls were prepared similarly, without adding the OEE1 culture filtrate. Slides were incubated in humid conditions at 25 °C, for 4 days. One slide from each concentration was observed daily under a microscope, to calculate the conidia germination and evaluate the eventual shape modifications.

### 2.7. Antifungal Traits of B. velezensis Strain OEE1

#### 2.7.1. Detection of Protease Production

Protease activity was revealed using two methods. Strain OEE1 was inoculated in a nutrient agar medium containing 100 g.L^−1^ skimmed milk, to evaluate the casein hydrolysis ability [22]. The second method was based on gelatin hydrolysis and consisted in inoculation of the bacterium in the same medium, containing 12 g.L^−1^ bacteriological gelatin. After incubation, the plates were covered with a mercuric chloride reagent [17]. The presence of a clear halo around the colonies designated enzyme production.

#### 2.7.2. Chitinase and β-glucanase Activity

Chitinase activity was carried out using a culture medium enriched in colloidal chitin, prepared according to Slama et al. [17]. Additionally, a barley flour agar plate was used to detect β-glucanase activity [17]. These activities were highlighted by the presence of clear halos around colonies.

#### 2.7.3. Hydrogen Cyanide (HCN) Production

Hydrogen cyanide (HCN) production was evaluated using the qualitative method of Slama et al. [17]. A fresh colony of the OEE1 strain was inoculated in 5 mL of an LB broth medium containing 4.4 g.L^−1^ glycine. Bands of filter paper pre-soaked in a solution of picric acid (2.5 g.L^−1^ picric acid and 12.5 g.L^−1^ Na_2_CO_3_), were inserted in the bacterial suspension. After incubation in a rotary shaker at 30 °C for 72 h, HCN production was marked by the filter paper colour’s change from yellow to brown/red.

### 2.8. Plant Growth Promotion (PGP) Traits of the B. velezensis Strain OEE1

#### 2.8.1. Siderophore Production

Iron chelating siderophore complexes production was evaluated using the chrom-azurol S (CAS) agar medium, as described by Slama et al. [17]. After 48 h of incubation at 30 °C, the apparition of halos around the colonies indicated siderophore production.

#### 2.8.2. Production of Indole-3-Acetic Acid (IAA)

The procedure described by Slama et al. [17] was used for the detection of IAA production in the LB broth medium, in the presence or absence of tryptophan. Approximately 4 mL of Salkowski’s reagent with two drops of the orthophosphoric acid were added to 1 mL of a fresh bacterial culture’s supernatant. After incubation (25 min at 30 °C in the dark), production of IAA was ascertained by the development of a characteristic pink colour.

#### 2.8.3. Phosphate Solubilisation

For assessment of phosphate solubilisation activity of the bacterial strains, we followed the protocol suggested by Slama et al. [17]. Briefly, bacterial isolates were plated on the Pikovskaya medium and incubated at 30 °C for 7 days. Phosphatase enzyme production was then attested by the appearance of clear zones around the colonies of bacteria.

#### 2.8.4. Nitrogen Fixation

The growth aptitude of the strain OEE1 in a nitrogen-free medium [17] indicates its ability to fix atmospheric nitrogen.

### 2.9. Endophytic Traits of B. velezensis Strain OEE1

#### 2.9.1. Cellulase Activity

Cellulase activity of the OEE1 strain was detected according to the procedure described by Slama et al. [17]. The OEE1 strain was, therefore, cultured on carboxymethyl cellulose (CMC) agar (8 days at 30 °C). Putative cellulase activity was then detected by overlaying the bacterial culture plates for 30 min, with a solution of 0.1% (w/v) Congo red. The plates were then bleached using 1M NaCl solution.

#### 2.9.2. Pectinase Activity

*B. velezensis* strain OEE1 was inoculated on a nutrient agar medium supplemented with 0.5% of pectin. A solution of 2% hexadecyl trimethyl ammonium bromide (CTAB) was then used to overlay the bacterial culture plates (30 min), followed by subsequent bleaching using a 1 M NaCl solution [17], in order to reveal pectinase activity.

#### 2.9.3. Amylase Activity

Production of α-amylase activity was assessed using the starch agar medium. After incubation for 48 h at 30 °C, hydrolysis zones were revealed after flooding the plates with iodine solution and an appearance of clear halos [17].

### 2.10. Samples Preparation for GC-MS and LC-HRMS Analysis

All reagents required for the analyses, such as formic acid, MS grade acetonitrile and water were purchased from Biosolve BV (Valkenswaard, the Netherlands). Bacterial fermentation was performed by inoculation of the OEE1 strain single colony in 50 mL of an ISP2 medium and incubation in a rotary shaker (25 °C, 180 rpm, 7 days). Diaion HP20 resin (50 g.L^−1^) was then incorporated in the flask, at the end of the fermentation procedure. After 6 h shaking, the mixture was centrifuged for 5 min at 10,000 rpm. The combined HP20 resin and call mass was extracted twice with methanol and the combined methanolic extracts was evaporated under vacuum, to obtain the residue that was used in the subsequent steps. For the GC-MS analysis, 10 mg of the methanolic extract residue was dissolved in methanol (10 mL) and fractioned in a separating funnel using *n*-hexane (2 × 10 mL). The hexane fraction was evaporated and 1 mg of the residue was re-dissolved in 10 mL of hexane. About 1 mL of the resulting solution was filtered through 0.2-µm PTFE filter, placed into the HPLC vial and submitted to a GC-MS analysis. For the LC-HRMS analysis, 1 mg of the obtained methanolic extract was re-dissolved in 10 mL methanol, filtered through 0.2-µm PTFE filter and dispensed in a HPLC vial, which was submitted for LC-HRMS analysis.

### 2.11. GC-MS Analysis

A Thermo HP-5MS column (30 m × 250 µm × 0.25 µm) purchased from J & W Scientific (USA) was attached to an Agilent 7820A gas chromatograph—operating on EI mode Agilent technologies 5975 series quadrupole mass spectrometer—and was used to analyse the volatile compounds in the bacterial strain OEE1. After injection of 1 µL sample and heating at 260 °C, the compounds were desorbed, prior to the programmed elution temperature. The program consisted of 50 °C for 5 min, then 50–250 °C at 5.7 °C/min. The carrier gas being helium (1.2 mL.min^−1^), 280 °C was the interface temperature for GC-MS analysis and the Agilent ChemStation software. NIST 11 Mass Spectral Library was used to tentatively identify compounds represented by peaks in the GC trace.

### 2.12. LC-HRMS Analysis: LC-HRMS Instrumentation and Conditions

A Thermo Scientific LTQ Orbitrap coupled to a Thermo HPLC system (PDA detector, PDA autosampler, and pump) were used to recover high resolution electrospray ionization mass spectra. The system was operated under the following conditions—capillary voltage and temperature set at 45 V and 260 °C, respectively; sheath gas flow rate and auxiliary gas flow rate set up at 40−50 arbitrary units and 10–20 arbitrary units, respectively; spray voltage of 4.5 kV; and a mass range adjusted to 100–2,000 amu (maximal resolution of 30,000). The HPLC column used for the LC/MS analysis was a Sunfire C18 analytical HPLC column (5 μm, 4.6 mm × 150 mm) operated over 30 min at a flow rate of 1 mL.min^−1^ using a mobile phase of 0 to 100% MeOH. Data was analysed using the Thermo Xcalibur 3.0 software and the dictionary of natural products database V. 23.1 (on DVD) was used for dereplication.

### 2.13. Bacterial DNA Extraction and Amplification

DNA from TSB-grown pure bacterial cultures was extracted using the Microbial DNA Isolation Kit UltraClean^®^ (QIAGEN, Basel, Switzerland), following the manufacturer’s guidelines. Agarose gel (1.5%) electrophoresis was used for visual assessment of DNA integrity [19]. Qubit Fluorometric Quantitation (Thermofisher, Switzerland) allowed for qualitative and quantitative assessment of the extracted DNA.

### 2.14. Bacterial Genome Sequencing Assembly and Annotation

*B. velezensis* strain OEE1 genome was sequenced using facilities available at the University of Geneva iGE3 genomics platform (http://www.ige3.unige.ch/genomics-platform.php). Using Illumina’s TruSeq sample preparation reagents, the sequencing library was prepared from gDNA of strain OEE1 and was processed according to Alenezi et al. [27,28]. After low quality reads filtering and de novo assembly of *B. velezensis* OEE1, the resulting contigs were deposited in the NCBI genome sequence database (accession no. MZXS00000000.1).

### 2.15. Selection of Genomes for Phylogenomic Analysis

Genomes of the different *B. velezensis* strains selected from among the genomes submitted to the NCBI genome sequence database for phylogenomic analysis are listed in Appendix A. Genome FASTA files as well as the amino acid sequences of the different strains were retrieved from the NCBI genome sequence database and was used in subsequent analyses.

### 2.16. Whole Genome Phylogeny

The software reference sequence alignment-based phylogeny builder—available at http://realphy.unibas.ch and thoroughly described in Bertels et al. [29]—was used to generate the genome alignments. Evolutionary distances were computed using the Kimura’s 2 parameter model and was used to generate the phylogenomic tree by the neighbour-joining (NJ) method, using the MEGA 6 program [30,31,32]. In the generated trees, bootstrap re-sampling support based on the 1,000 replications of the original data sets allowed the evaluation of branch validity.

### 2.17. Average Nucleotide Identity (ANI) Analysis

The algorithm developed by Goris et al. [33] was used to estimate the average nucleotide identity (ANI) values of the *B. velezensis* strains collection. Species boundary cut-off (set at 95~96%) by Richter and Rossello-Mora [34] was implemented in this software, available at the server EzBioCloud (http://www.ezbiocloud.net/tools/ani [35]) and was used in the current study.

### 2.18. Genome-to-Genome Distance Calculator (GGDC) Analysis

In silico genome-to-genome distance values of the *B. velezensis* strains collection was assessed using the web-based DSMZ (Deutsche Sammlung von Mikroorganismen und Zellkulturen) program available at http://ggdc.dsmz.de [36]. Species and sub-species cut-off suggested by default parameter analysis of the software (70%) were applied in the current study.

### 2.19. Comparative Genomics Analysis

Comparative genomics analysis of the *B. velezensis* strains collection was conducted exactly according to Belbahri et al. [19]. Mined features targeted those allowing bacterial fitness (nutrient acquisition, PGPR ability, root colonization, growth promotion factors, antibiotics and related compounds, resistance to drugs and heavy metals and degradation of aromatic compounds) and plant fitness (growth promoting traits (hormones), protection from oxidative stress and induction of disease resistance. A sequence identity cut-off of 50% was used in the BPGA (a Bacterial Pan Genome Analysis pipeline) pipeline [37] to compute the pan- and core-genomes of the *B. velezensis* strains collection (Appendix A). The USEARCH program implemented in the BPGA pipeline against the standard COG database allowed the assignment of core and pan-genomes’ functional genes into COG categories (https://iicb.res.in/bpga/index.html). BlastKOALA analysis of the *B. velezensis* genomes-predicted proteins allowed functional annotation, based on KEGG Orthology (KO), as suggested by Kanehisa et al. [38].

### 2.20. Identification of Core Genome and Accessory Genomes of the Isolate Collection

*B. velezensis* core genome (defined as sequences that are present in approximately all genomes from the isolates) and accessory genomes (defined as sequences that they are present only in some isolates of the collection) were determined using Spine and Agent, respectively [39].

### 2.21. B. velezensis Strain OEE1 Application and *F. solani* Infection of Olive Trees under Greenhouse Conditions

#### 2.21.1. Biocontrol Assay

Plant roots were carefully washed under running tap water to eliminate soil and was then soaked for 1 h in the *F. solani* conidial suspension. One week later, the roots were immersed into a bacterial suspension for curative treatments and, inversely, for protective treatments. A commercial fungicide named Uniform^®^, a mixture of Azoxystrobine and Mefenoxam molecules (Az + Mf) was used to compare the treatments’ efficacy. Plants inoculated with distilled water, fungal suspension and bacterial suspension were used as a negative control, a positive control and a bacterial control, respectively. The latter served to verify the absence of undesirable effects on olive trees, such as pathogenicity or phytotoxicity, and to survey the beneficial traits of strain OEE1. The experiment was conducted according to a randomized complete block with 10 plants for each, treatment and control plants. The inoculated plants were potted in polyethylene pots containing a sterile mixture of 50% peat and 50% sand. *Fusarium* wilt severity was estimated according to necrosis appearance on foliage, using the leaves damage index of a 0–4 scale (0 = 0%, 1 = 1–33%, 2 = 34–66%, 3 = 67–100%, 4 = dead plant). At the end of the experiment, pieces from the controls and the treated plants roots were plated in a PDA medium, for pathogen re-isolation, to fulfil the Koch postulates.

#### 2.21.2. Plant Growth Measurement

Growth promotion was analysed in 10 plants treated by the OEE1 spores and was maintained as described above. Plants length, collars diameter and the number of new shoots were determined weekly and were compared with the untreated plants.

#### 2.21.3. Colonization Assays

To evaluate the ability to colonize olive plants, roots were dipped in a suspension of the mutant bacterium OEE1, with a final concentration adjusted to 10^5^ CFU/mL. To confirm its migration and colonization in different vegetative tissues, re-isolation was assessed after 2 weeks. The leaves and stems were collected and sterilized using a 3% (v/v) sodium hypochlorite solution for 10 min and were subsequently washed with sterile distilled water, three times. Leaves and plugs from the stems were placed into a selective LB agar medium, amended with 60 mg/mL rifampicin and 100 mg/mL streptomycin and were previously covered by 100 µL of Fso1 conidial suspension. Presence of inhibition zones around vegetative tissues confirmed the endophytic colonization ability of the strain OEE1.

### 2.22. Statistical Analysis

One-way analysis of variance (ANOVA) and independent-samples T test implemented in IBM SPSS statistics software v. 22 was used to analyse the data. A post-hoc Tukey’s HSD test was also applied to compare the groups once significant effects were detected. The level of significance for all statistical tests was set to 5% (*p* < 0.05).

## 3. Results

### 3.1. Isolation of Endophytic Bacteria and Screening for Antifungal Activity

A total of 42 bacterial strains were isolated from olive trees cv. Chemlali, a widely cultivated variety in this area [40] that is highly sensitive to the majority of olive tree diseases [41]. All these strains were screened for their antagonistic activity using the dual culture method and 45% of them showed more than 60% mycelial growth inhibition (Appendix A). Interestingly, the root-derived strain OEE1 was able to inhibit *F. solani* growth by 82.42%. Thus, this strain was selected to investigate its antifungal potential against *F. solani* and its plant growth promoting ability.

### 3.2. Inhibition of Plant Pathogens by the B. velezensis Strain OEE1

The antagonistic potential of *B. velezensis* OEE1 was evaluated, based on the assessment of the size of the inhibition zone diameter in dual culture tests, with the examined fungi grown on PDA plates. Strain OEE1 showed varying degrees of inhibition of the examined pathogens. Percentage of inhibition ranged from 40–75% with oomycetes, including *P. ramorum*, *P. cactorum*, *P. cryptogea*, *P. plurivora* and *P. rosacearum*. It exhibited a lower effect on *Pythium* spp. and *Phytopythium* spp. than the *Phytophthora* spp., ranging from 25–40% against *P. sylvaticum*, *P. ultimum* and *P. vexans*. Moreover, strain OEE1 proved to be effective against *Fusarium avenaceum*, *F. sulphureum*, *Cladosporium cladosporioides* and *Botrytis cinerea*, with inhibition ranging from 40–60% (Figure 1a–c).

### 3.3. Antifungal Traits of B. velezensis Strain OEE1

Using the *F. solani* strain Fso1 in dual culture assays, the inhibitory effect of *B. velezensis* OEE1 was found to be effective and reached 82% (Figure 2). Microscopic observation of mycelial edges between this fungus and the OEE1 bacterium showed a strong cytoplasm vacuolization (Figure 2b1) and mycelial lysis (Figure 2b3). These phenomena were absent in the microscopic observations of the control plates (Figure 2b1).

Different concentrations of the cell-free culture filtrate were then used to check the antifungal activity of the secreted extracellular metabolites. A total inhibition of mycelial growth was reached with 75% metabolite solution, while 25% and 50% concentrations were only able to decrease the growth to up to half (Figure 2c). To check the fungicidal ability of these extracellular metabolites, discs dipped in the 75% metabolite solution were transferred to new PDA plates. Interestingly, after 48 h of incubation, no mycelial growth was noticed.

These experiments were followed by the evaluation of extracellular metabolites on conidia germination. Conidia of the *F. solani* strain Fso1 were cultured for 6 days on sterile slides containing a PDA medium mixed with 25%, 50% and 75% cell-free culture filtrate (Figure 2d). After 24 h of incubation, microscopic observation of these cultures showed a total inhibition of conidial germination, without shape variation, compared to the control, where about 22% of conidia started the germination process. Six days after incubation, up to 70% of conidia in the control slides gave rise to germination tubes (Figure 2d1), while no germination was noticed using 25% to 75% concentrations. A metabolite concentration of 75% was able to produce conidial lysis (Figure 2d3).

### 3.4. B. velezensis Strain OEE1 Plant Growth Promoting Traits

Plant growth promoting traits such as phosphate solubilisation, IAA biosynthesis, siderophore production and nitrogen-fixation were summarized in Appendix A. Strain OEE1 was able to solubilize phosphate, produce siderophore and IAA, even without any L-tryptophan induction. The bacterium was also able to grow in a nitrogen-free medium, suggesting its ability to fix atmospheric nitrogen.

### 3.5. Bacillus velezensis Strain OEE1 Endophytic Traits

The qualitative plate assay of hydrolytic enzymes activities showed the production of cellulase, pectinase and amylase by strain OEE1, as indicated by the clear zones of hydrolysis on CMC, pectin and starch agar media, respectively (Appendix A).

### 3.6. Phylogenetic Affinities of B. velezensis Strain OEE1

Genomes of 69 strains of *B. velezensis* available in GenBank (Appendix A) were retrieved and used in a phylogenomic approach to infer the exact phylogenomic position of strain OEE1. The genomic size of *B. velezensis* OEE1 was in the range of 4.07 Mbp (Appendix A). Four different species (represented by 22, 18, 14 and 15 strains each) lumped under the denomination *B. velezensis* were discovered using GGDC analysis sensu Meier-Kolthoff et al. [36]. The cut-off of 70% similarity between two genomes was suggested by these authors and set as a suitable cut-off and the gold standard threshold for species boundaries (Figure 3). ANI analysis agreed with the above results of GGDC analysis and four different species sensu Richter and Rosselló-Móra [34] were also confirmed. Cut-off (95–96%) set up to delimit species boundaries was unambiguously documented. Whole genome phylogeny also confirmed four sister clades in the tree, in total agreement with GGDC and ANI analysis (Appendix A). Strain OEE1 clustered within 15 isolates in a sister clade to the *B. velezensis* type strain KCTC 13012 [42].

### 3.7. GC-MS Analysis of the B. velezensis Strain OEE1

GC-MS analysis of the major VOCs of the *B. velezensis* strain OEE1 indicated the presence of phenylacetic acid, which is recognized as a member of the plant hormone class auxins and cyclo(Phe-Pro), which modulates auxin signaling in plants [43]. The intercellular signal molecule, indole, reported to regulate various aspects of bacterial physiology, including plasmid stability, spore formation, biofilm formation, resistance to drugs, and virulence was also detected. Ethylbenzene, phenylethyl alcohol, *E*-caryophyllene and cyclo(Leu-Pro) with pesticidal, antimicrobial, acaricidal and antifungal activities, respectively, were also detected in the VOCs of *B. velezensis* strain OEE1. Finally, numerous molecules with no known biological activities were also identified (Table 1).

### 3.8. LC-HRMS Analysis of B. velezensis Strain OEE1

LC-HRMS analysis of the *B. velezensis* strain OEE1 extract allowed the discovery of an impressive arsenal of surfactins, a type of biosurfactants exhibiting strong antibacterial and haemolytic activities [44]. Plipastatin B1 as well as several fengycins, including (Fengycin B, IX and XII), composed of ten amino acids linked to a C_14_–C_18_ β-OH fatty acid, known as a potent antifungal metabolite that inhibits filamentous fungi were identified in the *B. velezensis* strain OEE1 extract (Table 2).

### 3.9. Bioinformatic Evaluation of PGP Potential of B. velezensis Strains

Genes contributing to the plant beneficial functions were mined by homology to existing members in the *B. velezensis* strain OEE1 and the other 68 strain genomes, in order to in silico evaluate their plant growth promotion potential. Figure 4 shows the presence of large number of genes with plant beneficial functions independent of the coverage of the genome sequenced. There was no difference of the profile of the four putative species present in the species complex *B. velezensis*.

### 3.10. Secondary Metabolite Clusters

AntiSMASH 3.0, natural product domain seeker NapDos, prediction informatics for secondary metabolomes (PRISM), the bacteriocin-specific software BAGEL3 and NP.search [45,46,47,48,49] were used to mine the genomes of the 69 *B. velezensis* strains for the presence of putative secondary metabolite clusters. Diverse secondary metabolite clusters were uncovered from all genomes analysed, using different programs (Figure 5; Appendix A). Our analysis proved effective in recovering pathways for known secondary metabolites such as bacillomycin, amylocyclin, mersacidin, bacilysin, macrolactin, bacillibactin, bacillaene, surfactin, fengycin, difficidin, subtilin and locillomycin (Figure 5). However, high number of biosynthetic gene clusters failed to match pathways for known secondary metabolites, suggesting they are new and yet to be discovered. Rarefaction analysis of secondary metabolites pathways resulting from ongoing genome sequencing programs clearly attested that saturation was far from being achieved (Appendix A). The number of known gene clusters responsible for secondary metabolites biosynthesis detected using antiSMASH was not correlated with genome size. The genome size is able to explain up to approximately 0.9% of the variance in the number of secondary metabolites clusters (Appendix A). However, in the case of PRISM, genome size was able to explain up to 4.3% of the variance in the number of secondary metabolite clusters (Appendix A).

### 3.11. Bacillus Velezensis Genomes Prediction of Natural Products Richness and Location

Except for amylocyclin and bacillaene, which were harboured by the core genome, all other known secondary metabolites and the majority of the unknown ones were harboured by the accessory genome (Figure 6; Appendix A). The correlation between the accessory genome size and the number of gene clusters encoding secondary metabolites detected using antiSMASH was found to be weak (Appendix A). Only 2.3% of the variance in the number of secondary metabolites could be interpreted by genome size (Appendix A).

### 3.12. Characterization of B. velezensis Core and Pan Genomes

Belbahri et al. [19] suggested that full metabolic potentialities of a given species can be efficiently explored by pan and core genome analysis. Therefore, the core and pan genomes of the 69 isolates of *B. velezensis* available were deciphered and presented in Figure 8A,B. According to Figure 7A, the more genomes analysed, the more the pan genome of *B. velezensis* expanded. The increase in the pan genome size was set at the value of 0.16, which suggested that the species might have an open pan genome, according to Heaps’ law (as implemented in Tettelin et al. [60]). Using the concatenated amino acid sequences of the collection core genome in the exploration of their phylogenetic relationships, allowed to ascertain their species identity as *B. velezensis* (Figure 7C). GGD, ANI and phylogenetic tree analysis were all in perfect agreement that all studied isolates represented the *B. velezensis* strains.

### 3.13. Functional Characterization of the Core, Accessory and Unique Genomes of B. velezensis

The pan genome of a given species represents all known genetic variation for the species and, therefore, could be considered as a measure of the full genetic and metabolic potentialities of the species. Core genome is, however, the set of genes conserved across all genomes of the given species and, therefore, mirrors the conserved functional features of the species [19]. In order to assess these features in *B. velezensis* core and pan genomes, COG and KEGG distributions were assessed across all *B. velezensis* available genomes (Figure 9A,B). According to the COG distributions (Figure 8A), clear differences could be observed between core, accessory and unique genomes of the species. Unique genome, for example, was particularly rich in genes related to defense mechanisms (V), while transport, and catabolism (Q) and secondary metabolite biosynthesis were strongly represented in the accessory genome. Coenzyme transport and metabolism (H) as well as energy production and conversion (C) were, however, enriched in the core genome.

KEGG pathway analysis showed clear prevalence of xenobiotics degradation, membrane transport and signal transduction, metabolism of terpenoids and polyketides, as well as lipid, carbohydrate and amino acid metabolism functional genes in the accessory genome. This contrasted with the accumulation of energy and nucleotide metabolism and replication and repair of functional genes in the unique genome. Additionally, cell motility, translation and metabolism of cofactors and vitamins were particularly represented in the core genome of *B. velezensis*.

### 3.14. Application of B. velezensis Strain OEE1 under Greenhouse Conditions

The ability of strain OEE1 to cope with the incidence and severity of olive trees dieback and fine roots damage caused by the *F. solani* strain Fso1 was confirmed using 2-years-old olive trees cv. Chemlali, under greenhouse conditions. During the experimental period, the treated plants with OEE1 spores remained healthy. The infection rate and reduction of the disease severity were significantly modulated by the OEE1, as compared to the control groups (*p* < 0.05).

Disease indexes of plants treated by OEE1 were about 0.3 and 1 for the protective and curative treatments, respectively. Chemical fungicide was less effective in reducing disease severity, compared to the OEE1 treatment where disease indices were about 1.6 and 2 for the protective and curative treatments, respectively, compared to the severely infected positive control (Figure 9A). At the end of the experiment, the plants were extracted from plastic bags to investigate the root damages after treatments. While the roots were severely damaged in positive control and in the chemical-fungicide-treated plants in both curative and protective treatments, all other plant roots were well developed (Figure 9).

Treatment with the *B. velezensis* strain OEE1 positively influenced plant growth. The total number of leaves was increased, reaching at least 20 leaves, while it did not exceed 14 leaves in all other treatments. Apical elongation was also significantly enhanced (*p* < 0.05) and reached 80 mm compared to the negative control, where it was only about 40 mm. The same results were obtained for new shoots apparition and collar diameter evolution where the treatment with the *B. velezensis* strain OEE1 was extremely beneficial (Figure 9B).

### 3.15. Endophytic Colonization of B. velezensis Strain OEE1 of Olive Trees Cultivar Chemlali Roots

Investigation of the endophytic colonization of olive trees cultivar Chemlali roots by *B. velezensis* strain OEE1 was established using a mutant OEE1 strain designated OEE1’. The re-isolation of the bacterium from leaves (Figure 9c1,2) and stems (Figure 9c3,4) in plates amended with 60 mg/mL rifampicin and 100 mg/mL streptomycin and previously inoculated with the pathogen Fso1, showed the presence of impressive inhibition zones.

## 4. Discussion

Oomycota form a unique phylogenetic lineage of fungus-like eukaryotic microorganisms. They are mainly composed of highly damaging pathogens for animals, plants, fishes, insects, crustaceans and various microorganisms [2,5,6]. The genus *Phytophthora* within Oomycota continue to be devastating and cause serious disease outbreaks to plants in nurseries and wild ecosystems worldwide [4,7,8]. Recently, there has been much concern about sudden oak death and its causing agent, *P. ramorum* [11]. The Iberian Peninsula *Quercus ilex* and *Quercus suber* are severely threatened by *P. cinnamomi* [12]. An aggressive soil-borne plant pathogen *P. plurivora* has a worldwide distribution and is promoting beech decline in Europe [63]. Therefore, developing effective treatment against *Phytophthora* spp. is relevant for two reasons; their economic and environmental damages and the diversity of the hosts that they affect [8]. *Fusarium* diseases are also a major actor driving plant yield loss and mycotoxin contamination [17]. For decades, the control of *Fusarium* wilt disease remained ineffective [17]. Due to the lack of efficacy of current treatments, there is an urgent need for alternative solutions. Chemical treatments, while being widely used in agriculture are more abandoned, given their negative impacts on the environment and the development of resistance in *Phytophthora* populations, such as what has been observed with metalaxyl [64]. Biological control is an emerging trend in treating *Phytophthora* spp., among other disease-causing agents [13,16,65]. Biological control offers an eco-friendly and environmentally safe approach to reduce disease incidence by plant pathogens [19]. Indeed, few biological agents have been examined against *Phytopthora* spp. [66] and *Fusarium* spp. [17]. Recently, considerable knowledge has been acquired for effective development of biological control agents, using habitat-adapted symbiosis strategy [67,68]. Since olive trees were subjected to *Phytophthora* and *Fusarium* attacks [69,70,71], we have applied the habitat-adapted symbiosis approach to allow olive tree bacterial endophytes with *Phytophthora* and *Fusarium* biocontrol abilities to be recovered from sites subjected to *Phytophthora* and *Fusarium* attacks. During our screening of numerous olive trees bacterial endophytes, the strain OEE1, which proved to be effective in counteracting numerous *Phytophthora* and fungal phytopathogens was recovered (Figure 1). Strain OEE1 proved effective against at least three main *Phytophthora* pathogens including *P. ramorum*, *P. cactorum* and *P. plurivora*. Strain OEE1 also proved effective against two *Pythium* species (*P. ultimum* and *P. intermedium*), as well as against *Phytopythium vexans* (Figure 1). VOCs of the OEE1 strain proved effective against *P. vexans* and fungal pathogens *Botrytis cinerea* and *Rhizoctonia solani*, indicating the importance of both secreted metabolites and VOCs as biocontrol agents in counteracting plant pathogens [20,72].

Additionally, *B. velezensis* OEE1 proved effective in the control of *Fusarium* wilt of olive trees. Its significance in biological control was confirmed through several in vitro assays targeting inhibition of mycelium growth and conidia germination. *B. velezensis* strain OEE1 had a strong antifungal activity against *F. solani* FsoIt showed total inhibition of mycelial growth and blocked conidial germination. Extracellular metabolites secreted by OEE1 were fungicidal for *F. solani* Fso1, which failed to restore its normal growth. Microscopic observations revealed many mycelial alterations and conidial lysis. This fact is a critical step in the fungus life cycle and a selection based on this criterion makes this antagonist an efficient biocontrol agent [21]. This antifungal activity could be attributed to the production of many secondary metabolites and Cell Wall Degrading Enzymes (CWDE) that inhibit the fungal growth and spread. OEE1 has versatile facilities to produce protease, chitinase and β-glucanase, which are known to be efficient weapons against fungi, such as chitin, glucan and proteins, which are present in the major constituents of the fungal cell wall [22].

The *B. velezensis* OEE1 strain also proved to be endowed with PGP traits, such as siderophore production, indole-3-acetic acid production (IAA), phosphate solubilization and nitrogen fixation, suggesting its utility as a biofertilizer, in addition to its biocontrol features.

Analysis of secondary metabolites of the OEE1 strain using LC-HRMS analysis, proved to be effective in detecting an impressive arsenal of surfactins (a type of biosurfactants with antibacterial and haemolytic activities [44]) plipastatin B1 and several fengycins (including Fengycin B, IX and XII), which are known to be potent antifungal metabolites that inhibit filamentous fungi [44]. GC-MS analysis of the volatile fraction of the strain OEE1 revealed the presence of phenylacetic acid, which is recognized as a member of the plant hormone class auxins and cyclo(Phe-Pro), which modulates auxin signaling in plants [43]. These results highlight the importance of the VOCs of strain OEE1 in promoting plant growth and development. These results agreed with those of Tahir et al. [73] and Sanchez-Lopez et al. [74]. The intercellular signal molecule, indole, regulating resistance to drugs, spore formation, biofilm formation and virulence and plasmid stability (among various other aspects of bacterial physiology) was also detected in the VOCs of OEE1. These findings suggest that VOCs are also a rich source of the bacterial factors necessary for a successful symbiosis between the host and the plant beneficial bacterium [19]. Ethylbenzene, phenylethyl alcohol, *E*-caryophyllene and cyclo(Leu-Pro) with pesticidal, antimicrobial, acaricidal and antifungal activities, respectively, were also detected, suggesting that OEE1 has some rich weapons against pests and pathogens. Fatty acids were also found in the VOCs of strain OEE1, such as palmitic acid methyl esters (Table 1). According to Liu et al. [75], fatty acids show an effective activity against plant pathogens. Finally, numerous molecules with no known biological activities have also been discovered and they need further investigation by specifically targeted studies. Phylogenomic analysis of strain OEE1 and other 68 strains of *B. velezensis* suggested that the species suffer taxonomic imprecision. At least four species were lumped under the species name *B. velezensis*, which support the claim that *B. velezensis* is actually a species complex, as suggested by Belbahri et al. [19] and Dunlap et al. [76]. Using GGDC, ANI and phylogenomic inference should be mandatory methods in future studies to avoid taxonomic imprecisions that could impact the practical use of *B. velezensis* strains as effective biocontrol agents. The congruence of GGDC, ANI and phylogenomic analyses in delimiting species within *B. velezensis*, in a similar manner to that of *B. amyloliquefaciens* [19], warrant the strict application of these tools in biocontrol bacterial taxonomy. Genome mining of the 69 strains of *B. velezensis*, using homology-based mining of genes contributing to plant-beneficial functions, revealed an impressive battery of genes that contribute to plant growth promotion that suggest the possibility of using this strain as a biofertilizer (Figure 4). Secondary metabolites predicted from the genomes of *B. velezensis* strains using prediction informatics for secondary metabolomes (PRISM) [47], antiSMASH 3.0 [45], NP.search [49], NapDos [46], and the bacteriocin-specific software BAGEL3 [48] proved to be very diverse (Figure 5, Appendix A). *B. velezensis* strain OEE1 had the second most impressive record of secondary metabolites described to date (Table 2). The high number of unknown secondary metabolite clusters warrant the development of targeted studies to discover their functions and their target organisms. These results agreed with those of Belbahri et al. [19] and Loper et al. [77]. Rarefaction analysis of secondary metabolite clusters, across genome sequencing, suggested that more efforts are required to describe hidden and unlocked secondary metabolite biosynthesis potential in *B. velezensis*, which further strengthens the findings of Belbahri et al. [19] and tends to generalize these findings to another species of the “operational group *Bacillus amyloliquefaciens*” [42]. No correlation between the number of secondary metabolites described and the genome size was found in this study, which was not the case in Belbahri et al. [19] for *B. amyloliquefaciens*. This finding suggested a specificity of genome architecture among the “operational group *B. amyloliquefaciens*”. Few secondary metabolites were found in the core genome of *B. velezensis* (Figure 6), which strengthen the findings of Belbahri et al. [19] and tend to generalize the importance of accessory genome as a backyard for secondary metabolite production across a given species strains. No correlation was observed between the accessory genome size and the number of discovered secondary metabolites pathways.

Effectiveness of *B. velezensis* OEE1 in controlling the dieback caused by the pathogens *F. solani* Fso1 was revealed by significantly lower disease indices in the in planta experiments using OEE1, either in curative or preventive treatments against *F. solani* Fso1. Moreover, *B. velezensis* OEE1 exhibited a biofertilizer agent behaviour along with the powerful biocontrol activity against *F. solani* Fso1. These results were revealed both with in vitro and *in vivo* experiments. Treatment with the strain OEE1 had significantly improved the growth of olive trees. The treated plants showed higher length and better vegetative and root development, compared to the rest of plants. These beneficial effects could be explained by the phosphate solubilisation, nitrogen fixation and IAA production [17]. These characters were provided by the OEE1 endophytic life style. Plant penetration is generally assisted by the ability to hydrolyse plant cell walls via hydrolytic enzymes found in OEE1 strains, such as pectinase, amylase and cellulase [17]. Gram-positive bacterium *B. velezensis* is known to be among the best producers of antimicrobial metabolites [19]. In conclusion, our findings strongly suggest that the strain OEE1 is an effective biocontrol agent that could offer not only plant protection against *Phytophthora* and other fungal diseases but also provide plant growth promotion. Taken together, the results of this study are in clear agreement with the habitat-adapted symbiosis ecological concept proposed by Rodriguez et al. [67,68], and suggest that this strategy could provide a direct approach to enrich biocontrol toolbox against recalcitrant *Phytophthora* and *Fusarium* species.

## Figures and Tables

**Figure 1 microorganisms-07-00314-f001:**
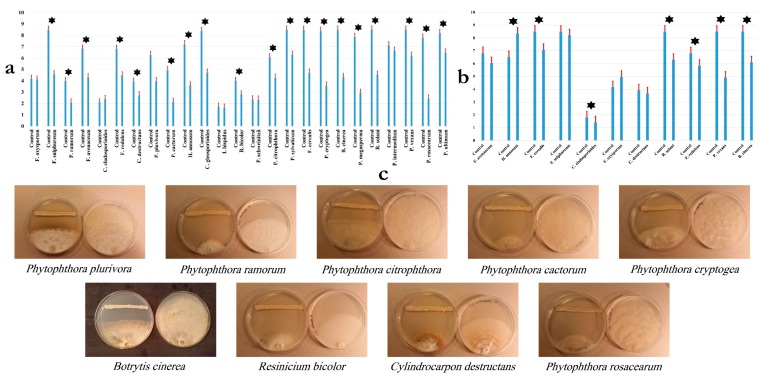
(**a**,**b**) Inhibition of *Bacillus velezensis* OEE1 against fungal species. Data presented as mean ± standard error. Bars labelled with the asterisks are significantly different from control according to Tukey’s HSD at *p* < 0.05. (**c**) Confrontation assay of antifungal activity of the bacterial isolate OEE1 against different fungal species. A representative control petri dish of each fungal species is presented.

**Figure 2 microorganisms-07-00314-f002:**
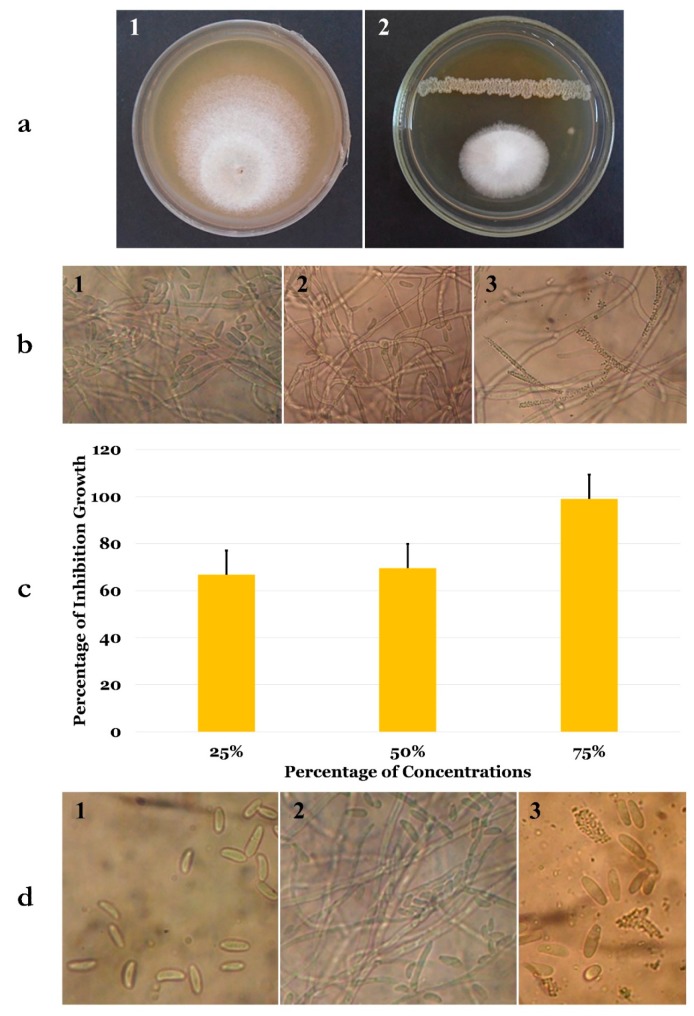
(**a**) Antifungal activity of bacterial isolate OEE1 against *Fusarium solani* after 6 days of incubation at 25 °C. (**b**) Microscopic examination (×400) of *F. solani* mycelia after confrontation assay with bacterial isolate OEE1 for 6 days at 25 °C (from left to right—negative control, cytoplasm vacuolization and mycelial lysis). (**c**) Inhibition percentages of different concentrations of the isolate OEE1 culture filtrate on *F. solani* mycelial growth at 25 °C after 6 days of incubation. (**d**) Microscopic examination (×400) of culture filtrate inhibition on conidia germination after growth at 25 °C (from left to right—*F. solani* conidia at the time of incubation, negative control after 6 days of incubation and conidial lysis after 6 days of incubation with a 75% culture filtrate).

**Figure 3 microorganisms-07-00314-f003:**
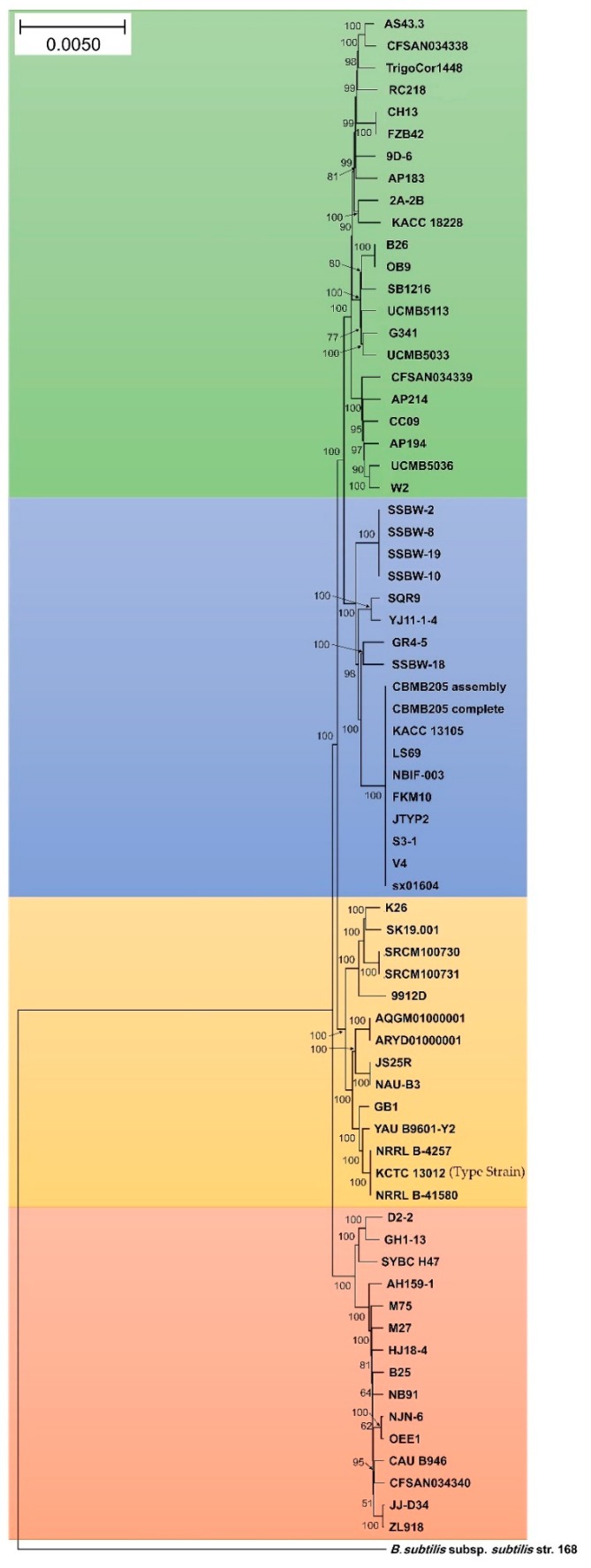
Neighbour-joining phylogenomic tree of Gram-positive bacteria *Bacillus velezensis* isolates. *Bacillus subtilis* subsp. *subtilis* isolate 168 was used as the outgroup. Supports for branches were assessed by bootstrap resampling of the data set with 1,000 replications. Each colour corresponds to a “sister clade” that represent a putative new species.

**Figure 4 microorganisms-07-00314-f004:**
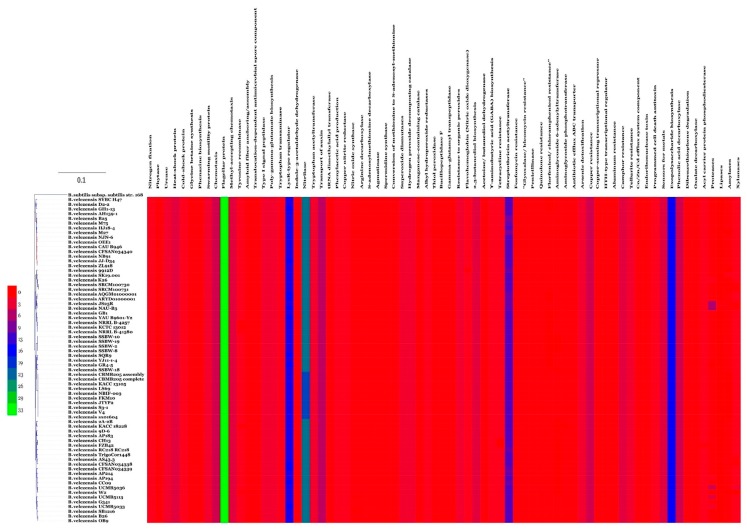
Heat map resulting from the genome mining of genes contributing to plant-beneficial functions in *B. velezensis* isolates. Gene copy number varies between 0 and 33 according to the indicated colour code.

**Figure 5 microorganisms-07-00314-f005:**
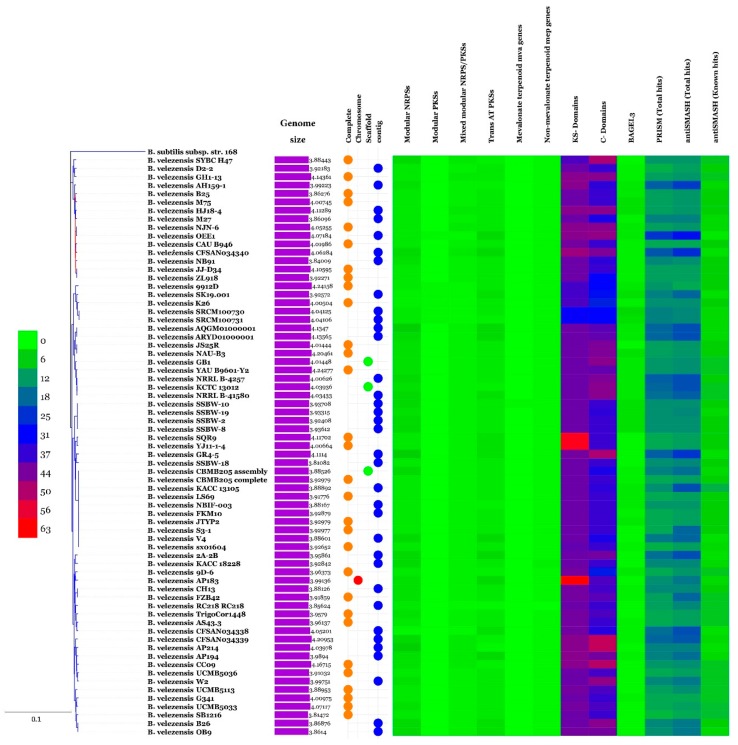
Genome size (MB), level of whole genome sequencing and a heat map resulting from the genome mining of genes contributing to secondary metabolite clusters. Gene copy number varies between 0 and 63, according to the indicated colour code.

**Figure 6 microorganisms-07-00314-f006:**
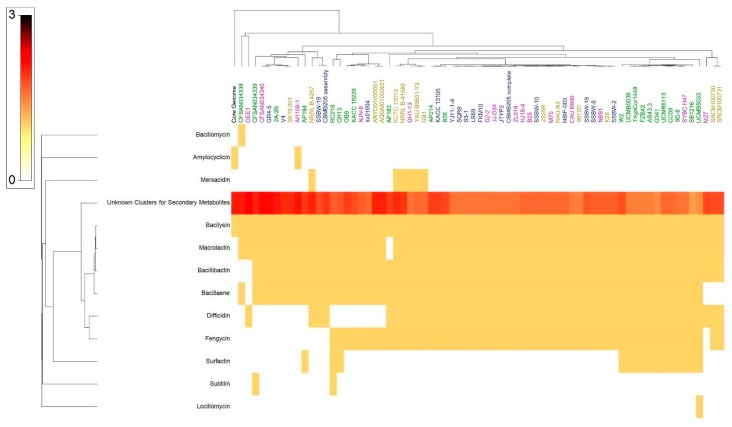
Heat map of *B. velezensis* accessory genome secondary metabolites clusters. The number of secondary metabolites clusters varied between 0 and 3.

**Figure 7 microorganisms-07-00314-f007:**
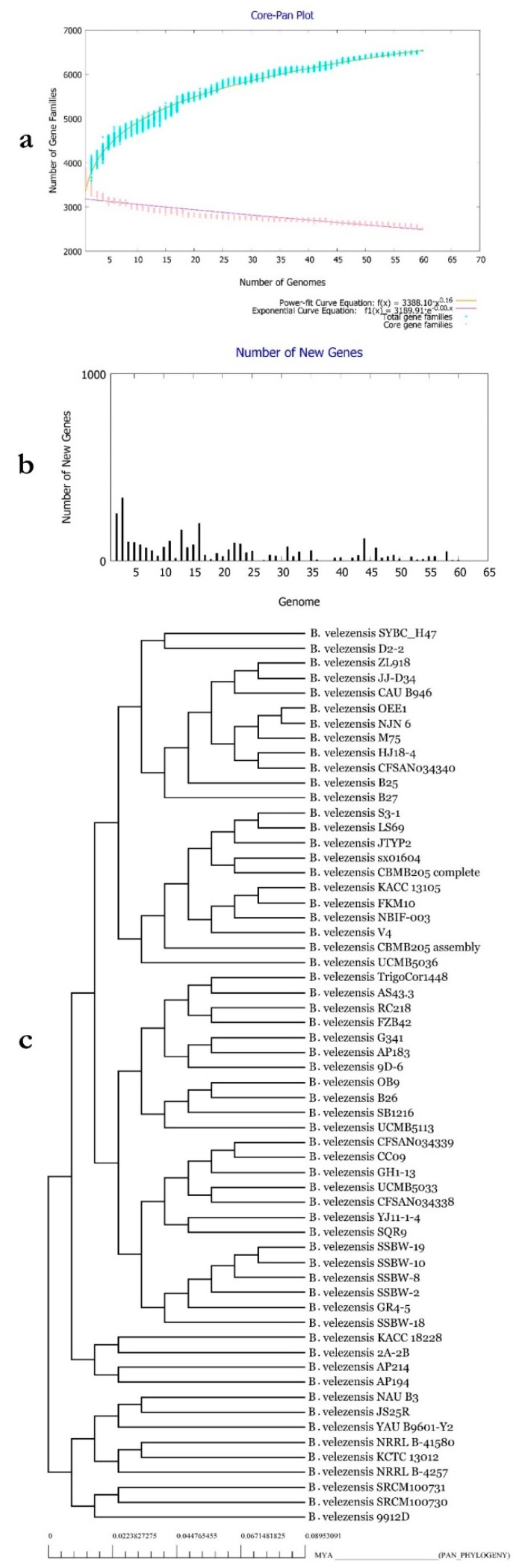
(**a**) Core-Pan genome plot of *B. velezensis* isolates based on the number of genomes and number of gene families. (**b**) Number of new genes identified in the genomes of the bacteria. (**c**) Tree of Pan-genome related phylogeny of *B. velezensis* isolates.

**Figure 8 microorganisms-07-00314-f008:**
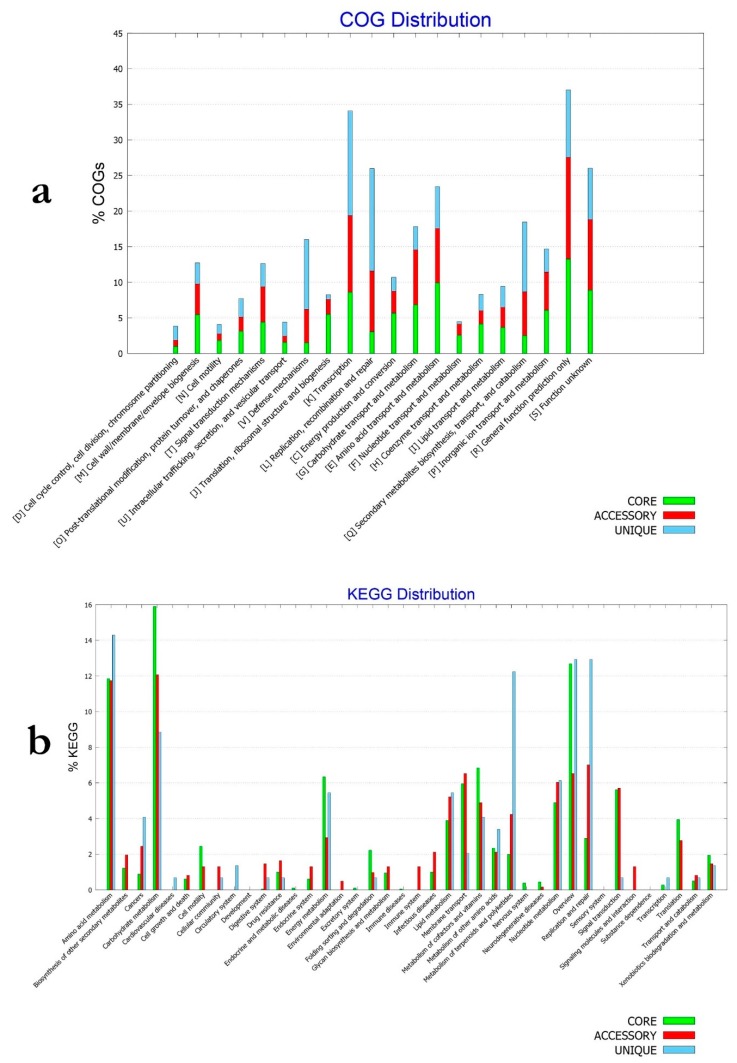
(**a**) COG and (**b**) KEGG distribution among core, accessory and unique genomes of *B. velezensis* isolates.

**Figure 9 microorganisms-07-00314-f009:**
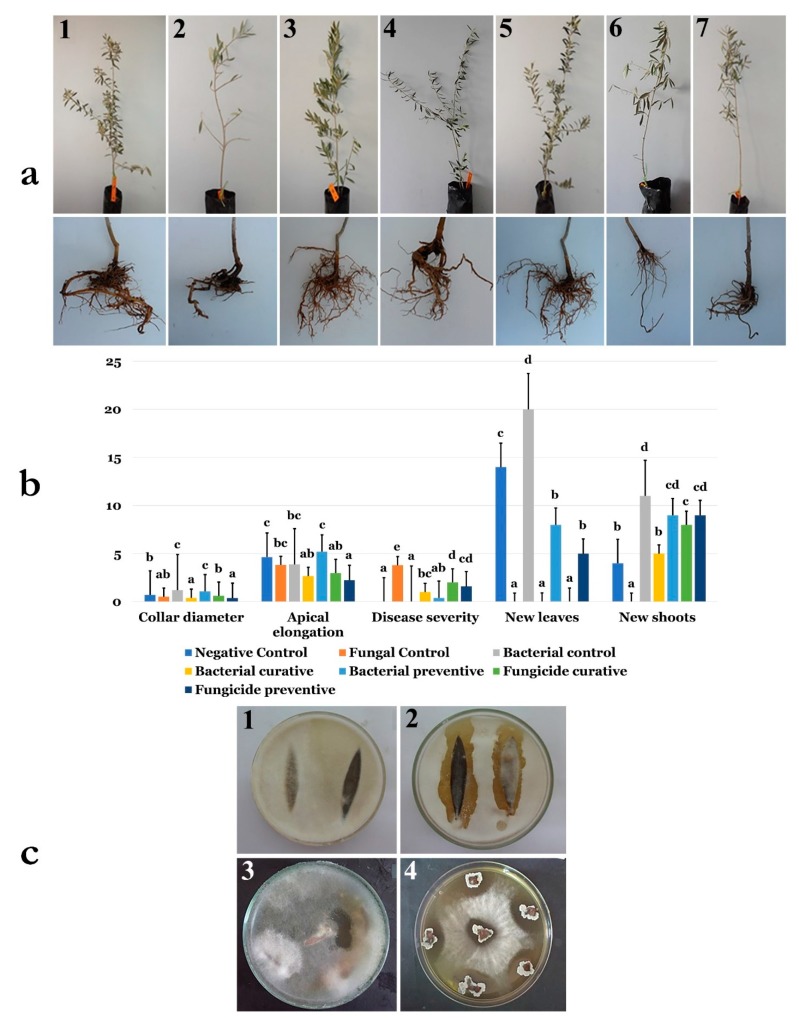
(**a**) Effect of preventive, concomitant, and curative biocontrol treatment on olive trees after 4 months post-inoculation with *F. solani* strain Fso1—representative non-treated (control) olive trees (1), infected with *F. solani* Fso1 (2), inoculated with *B. velezensis* OEE1 (3), infected with *F. solani* Fso1 and then treated with *B. velezensis* OEE1 (OEE1 curative treatment) (4), inoculated with *B. velezensis* OEE1 and then infected with *F. solani* Fso1 (OEE1 preventive treatment) (5), infected with *F. solani* Fso1 and then treated with the fungicide (fungicide curative treatment) (6) and inoculated with *B. velezensis* OEE1 and then treated with fungicide (fungicide preventive treatment) (7), respectively. (**b**) Effects of the different treatments (negative control, fungal control, bacterial control, bacterial curative, bacterial preventive, fungal curative and fungal preventive) on collar diameter, apical elongation, disease severity, number of new leaves and new shoots. Data presented as mean ± standard error. Bars labelled with different letters were significantly different among the treatments at *p* < 0.05, using the Tukey’s HSD test. (**c**) Re-isolation of the mutant strain OEE1′ from (1,2) leaves and (3,4) stems after root inoculation with *B. velezensis* OEE1′. The mutant strain was used in parts 1, 2, 3 and 4.

**Table 1 microorganisms-07-00314-t001:** Results retrieved from the GC-MS analysis of the *B. velezensis* strain OEE1.

Retention time (Rt)	Kovat’s Index	Lib Score	Tentative Identification	Plant Growth Promotion (PGP) Effectiveness or Biocontrol Ability	Reference
8.981	854	92.0	Ethylbenzene	Used as pesticides	[50]
11.473	1082	93.8	Phenylethyl Alcohol	Antimicrobial activity, an auto-antibiotic produced by the fungus *Candida albicans*	[51]
13.909	1174	84.6	Indole	As an intercellular signal molecule, indole regulates various aspects of bacterial physiology, including spore formation, plasmid stability, resistance to drugs, biofilm formation and virulence	[52]
16.368	1187	94.5	1-Dodecene		
18.656	1251	92.1	Benzene acetic acid (Phenylacetic acid)	An active auxin (a type of plant hormone), naturally produced by the metapleural gland of most ant species and used as an antimicrobial	
20.942	1337	82.1	Eugenol	Zinc oxide eugenol is used for root canal sealing	[53]
21.103	1424	69.8	E-Caryophyllene	Widely distributed among plant oils, and reportedly possess acaricidal, insecticidal, repellent, attractive and antifungal properties	[54]
22.116	1587	93.9	1-Hexadecene		
22.995	1478	82.1	β-Selinene (β-Eudesmene)	A sesquiterpene hydrocarbon with a naphthalene skeleton and is a component of celery oil. The autoxidation product of it has antimalarial activity	[55]
23.563	1496	79.8	Cyclo(Leu-Pro)	Strongly inhibit mycelia growth of fungus and thereby affecting aflatoxin production	[56]
24.498	1627	71.2	3-Hydroxy-β-damascone	Inducer of NAD(P)H:quinone reductase (QR) activity; novel inhibitors of inducible nitric oxide synthase (iNOS) induction	[57]
Rt	Kovat’s Index	Lib Score	Tentative Identification	PGP Effectiveness or Biocontrol Ability	Reference
25.458	1864	78.5	N-cetyl alcohol		
25.897	1891	92.1	1-Nonadecene		
26.172	1908	72.5	Palmitic acid methyl ester		
30.796	1942	83.2	Palmitic acid		
31.173	2070	79.3	1-Octadecanol (Stearyl alcohol)		
31.395	2073	93.6	Methyl linoleate		
32.924	2084	67.1	Methyl oleate	Identified as a primer pheromone in honeybees	[58]
33.687	2138	94.1	Cyclo(Phe-Pro)	A secondary metabolite produced by certain bacteria and fungi controls the expression of genes involved in pathogenicity, cell-to-cell communication by bacteria, It also modulates auxin signaling in plants.	[59]
39.015	2704	73.5	Diisooctyl phthalate		

**Table 2 microorganisms-07-00314-t002:** Results retrieved from the LC-MS analysis of the *B. velezensis* strain OEE1.

Retention time (Rt)	Tentative Identification	Formula of the Molecule	PGP Effectiveness or Biocontrol Ability	Reference
22.06	Surfactin B	C_52_H_91_N_7_O_13_	A heptapeptide linked with a C_13_–C_16_ β-OH hydroxyl fatty acid, is a type of biosurfactant with antibacterial and haemolytic activities, but it has no inhibitory effects on filamentous fungi.	[61]
22.73	Surfactin C15	C_53_H_93_N_7_O_13_		
14.88	Plipastatin B1	C_74_H_114_N_12_O_20_	Composed of ten amino acids linked to a C_14_–C_18_ β-OH fatty acid, is an antifungal metabolite that inhibits filamentous fungi, while it has no effect on yeasts and bacteria.	[62]
14.12	C_16_-Fengycin B	C_73_H_112_N_12_O_20_	Composed of ten amino acids linked to a C_14_–C_18_ β-OH fatty acid, is an antifungal metabolite that inhibits filamentous fungi, while it has no effect on yeasts and bacteria.	[62]
14.41	Fengycin IX	C_72_H_110_N_12_O_20_		
15.41	Fengycin XII	C_75_H_116_N_12_O_20_		

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
