# Peer review of "Olea europaea L. Root Endophyte Bacillus velezensis OEE1 Counteracts Oomycete and Fungal Harmful Pathogens and Harbours a Large Repertoire of Secreted and Volatile Metabolites and Beneficial Functional Genes"

_microorganisms, 2019, doi:10.3390/microorganisms7090314_

Round 1

Reviewer 1 Report

In this article the authors reported an in-depth characterization of a Bacillus velezensis strain. A phenotypic and genomic characterization of this strain has been carried out, highlighting the ability to both inhibit several plant pathogens and promote plant growth.

 Although the data reported in the article are potentially interesting and a great deal of work has been done, in my opinion the article cannot be published in the present form.

Several revisions are needed. The introduction is not clear in some points and throughout the article there are some formatting errors (for example the names of the strains in many cases are not in italic).

According to the introduction the “microbiome of olive trees under Phytophthora and fungal infection threat was use as a starting point for wide screening of root endophytes against large collection of Phytophthora and fungal pathogens”. But the procedure by which these endophytes were isolated and how and why the Bacillus velezensis strain was chosen among the others is not reported clearly.

The phenotypic data that should be reported in table 2 are missing (see my specific comments) and in general the order in which the results are reported is confusing. The quality of the figures is low, in many cases nothing can be read. A paragraph that should be in the materials and methods is found in the results.

My suggestion is to review the article, in particular the results, reviewing the order in which they are reported, summarizing them in a clearer way and defining better the purpose of each analysis.

 Table 1 and many of the figures could be moved to additional materials, leaving only a few figures in the text that are fundamental to understand the results. Materials and methods can also be summarized. Many of the methods used has been already published and could be reported only as additional files to shorten the manuscript and facilitate the reading.

Specific comments

Introduction:

Line 65-68: this sentence is not very clear

Line 74-76: not clear

Line 84: strain among species… what it means?

Materials and methods

Line 100: why use 37°C for environmental strains?

Line 112: Check the names of microorganisms in many cases they are not in italic

Line 120: The number of cells used has been verified in some way?

Line 172: HCN define the abbreviation

Line 232: remove the point, remove through filter.

The materials and methods for the core genome analysis are missing

Results:

Table 1: move to supplementary materials

Line 371-375: these results are not reported in table 2 as indicate by the text and I could not find these results anywhere in the manuscript

Line 377-379: same thing the results are missing

Figure 3B: the figure is absolutely incomprehensible, improve also the quality of figure 3C

The meaning of the two heatmaps is not clear. The numbers are the copies of genes in the genome for each category? Add a legend

In general, the exposure of the results is confusing. First, the authors report the phenotypic characterization of the strain, then they move to the genomic analysis, then they return to the characterization of the molecules produced, then back to the genomes. The description of the pangenoma is also confusing ... it is better to first describe the pangenoma in general and then go into details (Paragraph 3.12 should be placed before paragraph 3.11)

Figure 6: move to supplementary files

Line 454-459 : materials and methods

Figure 7B: move to supplementary files

Author Response

Dear Editor in Chief of Microorganisms,

We are grateful to the anonymous reviewers; their comments were highly valuable for providing the best relevant data to the journal readers. We believe this highly improved the quality of the manuscript. Therefore, we followed the vast majority of their suggestions and made the suggested changes in the manuscript. Below we provide a list of reviewers’ recommendations and the changes operated in the revision.

Reviewers' comments:

Academic Editor Notes

If the reviewers have suggested that your manuscript should undergo extensive English editing, please address this during revision. We suggest that you have your manuscript checked by a native English-speaking colleague or use a professional English editing service. Alternatively, MDPI provides an English editing service checking grammar, spelling, punctuation and some improvement of style where necessary for an additional charge (extensive re-writing is not included), see details at https://www.mdpi.com/authors/english.

Reviewer #1:

Comments and Suggestions for Authors

In this article the authors reported an in-depth characterization of a Bacillus velezensis strain. A phenotypic and genomic characterization of this strain has been carried out, highlighting the ability to both inhibit several plant pathogens and promote plant growth. Although the data reported in the article are potentially interesting and a great deal of work has been done, in my opinion the article cannot be published in the present form. Several revisions are needed.

Corresponding Author: According to reviewer recommendation the paper have been improved.

The introduction is not clear in some points and throughout the article there are some formatting errors (for example the names of the strains in many cases are not in italic).

Corresponding Author: Scientific names of the strain are in italic now. The introduction has also been clarified according to reviewer recommendation.

According to the introduction the “microbiome of olive trees under Phytophthora and fungal infection threat was use as a starting point for wide screening of root endophytes against large collection of Phytophthora and fungal pathogens”. But the procedure by which these endophytes were isolated and how and why the Bacillus velezensis strain was chosen among the others is not reported clearly.

Corresponding Author: The issue was resolved and the following text was inserted in the manuscript:

Isolation of endophytic bacteria and screening for antifungal activity

Isolation of bacterial endophytes was performed from healthy olive trees belong to the cultivar Chemlali located in Sfax (South Tunisia 34°43'50.7"N 10°44'08.6"E). The entire plant was washed under running water to remove soil and dust and endophytic bacteria were isolated from each plant part as described by [73]. To ensure the surface sterilization efficacy, 100 µL of rinsing water from the last washing was plated on TSA medium and incubated at 30 °C [74].

Antifungal activity of the isolated bacteria was evaluated using the dual culture test [17] against Rhizoctonia bataticola HQ392809.1, R. solani KU863546, Neofusicoccum australe EU375516.1, Nigrospora sp. JN 207298.1, Botryospheria sp., F. solani FJ874633.1, Pu: Pythium ultimum, F. oxysporum JN400698.1 and Cylindrocarpon sp. The isolate showing an ability to inhibit the most of pathogens and a strong activity against F. solani was selected for the rest of this study.

Results

A total of 42 bacterial strains were isolated from olive trees cv. Chemlali, a widely cultivated variety in this area [75] and highly sensitive to the majority of olive tree diseases [76]. All these strains were screened for their antagonistic activity using the dual culture method and 45 % of them showed more than 60% mycelial growth inhibition (Table S3). Interestingly, the root-derived strain OEE1 was able to reduce F. solani growth to 82.42%. Thus, this strain was selected to investigate its antifungal potential against F. solani and its plant growth promoting ability.

Table S3. Antifungal activity of endophytic bacteria against the olive trees pathogens using the dual culture method

Rb

Pu

Rs

Na

B

N

Fs

Fo

C

Control

0.00

0.00

0.00

0.00

0.00

0.00

0.00

0.00

0.00

10R3I

86,14 ± 0.72b

86,39 ± 0.26b

16,22 ± 1.70n

21.36 ± 0.02mn

63.32 ± 0.39f

25.89 ± 0.88m

60.24 ± 0.59c

54.27 ± 0.58g

70.35 ± 0.69c

11R3I

13,30  ± 2.25kl

20,98 ± 0.11l

20,89 ± 1.58m

33.83 ± 1.04k

0.00

0.00

29.46 ± 0.36h

0.00

0.00

12R3I

63,13 ± 2.24d

32,22 ± 0.45k

28,37 ± 1.73j

36.21 ± 0.95j

44.10 ± 2.54h

0.00

54.92 ± 0.41d

13.58 ± 0.65o

64.91 ± 0.32e

13R3I

26,50 ± 0.2h

89,03 ± 0.07a

50,44 ± 0.58h

74.03 ± 0.32bc

31.96 ± 1.37i

23.14 ± 0.31m

0.00

8.29 ± 1.28q

69.75 ± 1.20c

14R3I

0.00

20,98 ± 1.01l

35,92 ± 1.70i

28.23 ± 0.45l

0.00

0.00

12.57 ± 0.85m

0.00

4.32 ± 0.03r

15R3I

30,42 ± 0.91g

51,83 ± 0.21h

19,20 ± 1.02m

19.74 ± 0.09n

28.45 ± 0.21i

11.5 ± 0.67o

28.62 ± 1.22h

60.38 ± 0.31ef

0.00

17B1I

51,89 ± 2.88e

73,17 ± 0.39de

25,49 ± 0.91kl

32.51 ± 1.23k

82.37 ± 0.69bc

72.15 ± 1.33c

64.91 ± 0.09b

57.36 ± 0.97f

76.22 ± 0.52b

OEE1

61,09 ± 0.65d

56,47 ± 3.55fg

84,30 ± 1.15a

85.78 ± 0.42a

86.38 ± 0.54a

82.54 ± 0.25a

82.42 ± 0.15b

42.86 ± 0.64i

81.47 ± 0.47a

1B2I

0.00

45,10 ± 1.81i

0,00

0.88 ± 0.63s

0.00

0.00

0.00

13.82 ± 0.09o

4.61 ± 0.61r

1C3I

56,72 ± 1.01d

7,49 ± 0.23no

7,34 ± 1.30o

16.06 ± 0.33o

31.91 ± 2.14i

0.00

26.48 ± 0.37i

37.60 ± 0.22j

44.35 ± 0.32j

1F1I

0,00

9,00 ± 0.01n

0,00

0.00

0.00

0.00

4.36 ± 0.94o

10.21 ± 0.03p

0.00

1F2I

35,52 ± 0.44fg

39,41 ± 3.01j

45,17 ± 0.03hi

58.57 ± 0.41f

0.00

62.33 ± 0.89fg

2.58 ± 0.60p

46.24 ± 0.72h

61.75 ± 0.08fg

1R3I

79,78 ± 1c

26,43 ± 0.11k

68,93 ± 0.76e

69.11 ± 1.12c

83.20 ± 0.085b

64.38 ± 0.37e

46.21 ± 0.03f

70.59 ± 0.83c

47.21 ± 0.64hi

1T1I

20,81 ± 0.04j

61,78 ± 0.49f

69,81 ± 0.24d

72.65 ± 0.37c

8.51 ± 0.61m

35.26 ± 0.14jk

28.24 ± 0.77hi

66.35 ± 0.23d

48.91 ± 0.01h

20R3I

1,00 ± 1.73m

11,15 ± 0.34n

0,00

0.00

43.71 ± 0.12h

0.00

0.00

21.34 ± 0.94m

0.00

20R3I'

0,00

19,49 ± 1.43l

0,00

0.00

0.00

29.20 ± 2.97l

0.00

0.00

5.36 ± 1.24r

2B1F

23,10 ± 0.33i

15,55 ± 0.84m

2,52 ± 1.03p

6.46 ± 0.52q

32.18 ± 0.66i

2.31 ± 2.32q

73.26 ± 0.26a

0.00

67.92 ± 0.68d

2B1I

74,03 ± 0.57cd

27,74 ± 0.64k

66,78 ± 1.76f

69.91 ± 0.69c

68.92 ± 0.73e

78.31 ± 0.90ab

59.28 ± 0.42c

66.84 ± 0.56d

46.80 ± 0.07i

2B2I

0,00

3,00 ± 0.1o

0,00

0.00

0.00

24.59 ± 0.38m

0.00

0.00

0.00

2C3I

51,75 ± 2.31e

7,52 ± 0.15no

0,00

2.20 ± 0.35rs

54.91 ± 2.39g

46.2 ± 0.66i

27.13 ± 0.83i

0.00

33.56 ± 0.28l

2F1I

0,00

8,15 ± 0.56no

25,31 ± 1.03kl

24.11 ± 1.08m

31.96 ± 0.48i

19.67 ± 0.07n

20.61 ± 0.68k

0.00

10.82 ± 0.33p

2R3I

20,02 ± 2.31j

76,02 ± 0.06d

49,20 ± 1.45h

53.98 ± 0.94g

16.20 ± 0.90k

37.29 ± 0.58j

24.66 ± 0.49j

41.09 ± 0.38i

7.34 ± 0.19q

2T1I

0,00

87,81 ± 0.68a

24,28 ± 1.21l

36.90 ± 0.67j

12.50 ± 1.50l

0.00

39.88 ± 1.08g

60.75 ± 2.06ef

0.00

3B1I

58,45 ± 0.9d

14,25 ± 0.15m

69,54 ± 0.21de

69.66 ± 0.28c

81.60 ± 0.11c

76.14 ± 0.21b

38.09 ± 0.16g

57.92 ± 0.33f

69.83 ± 0.21c

3B2I

0,00

68,78 ± 1.74e

70,54 ± 1.82d

67.82 ± 0.84d

0.00

0.00

51.40 ± 0.73e

74.36 ± 0.49b

9.29 ± 0.43p

3R3I

5,46 ± 0.46lm

52,67 ± 3.04g

53,86 ± 0.85g

58.21 ± 0.12f

0.00

46.97 ± 0.96i

23.50 ± 0.89j

0.00

69.84 ± 0.97c

3T1I

11,27 ± 0.04l

20,16 ± 0.66l

17,49 ± 1.39n

38.99 ± 1.61i

0.00

18.97 ± 0.72n

24.58 ± 0.44j

3.21 ± 0.81r

0.00

4B1I

0,00

0,00

53,02 ± 0.61g

0.00

25.92 ± 0.82ij

0.00

0.00

44.67 ± 0.66h

0.00

4B2I

8,77 ± 0.03l

6,97 ± 1.15no

65,14 ± 1.33f

71.17 ± 0.24c

0.00

62.74 ± 0.02fg

0.00

18.63 ± 0.32mn

0.00

4C3I

14,13 ± 2.47k

61,14 ± 0.8f

0,97 ± 0.52pq

11.94 ± 0.61pq

19.82 ± 1.86j

0.00

0.00

31.57 ± 0.91k

16.80 ± 0.38n

4R3I

0,00

3,82 ± 0.05o

69,93 ± 0.48d

70.02 ± 1.33c

0.00

49.65 ± 0.34h

8.03 ± 0.04n

0.00

58.28 ± 0.45g

5B2I

0,00

3,91 ± 0.6o

0,00

57.44 ± 0.92

21.69 ± 0.49j

37.19 ± 1.69j

0.00

0.00

22.64 ± 0.97m

5R3I

42,88 ± 0.94f

7,27 ± 0.81no

2,30 ± 1.00p

12.6 ± 0.09p

0.00

0.00

9.38 ± 0.13n

59.01 ± 0.64f

0.00

6B1I

0,00

31,56 ± 4.1k

0,86 ± 1.49pq

8.14 ± 0.25q

22.73 ± 0.72j

6.01 ± 0.72p

0.94 ± 0.81q

0.00

0.00

6B2I

20,88 ± 0.16j

82,78 ± 0.83c

41,74 ± 0.85hi

45.58 ± 0.86h

0.00

0.00

46.85 ± 0.60f

17.91 ± 0.82n

22.19 ± 0.83m

6R3I

42,05 ± 0.89f

75,60 ± 0.04d

73,39 ± 0.42c

75.89 ± 2.01bc

0.00

51.48 ± 0.11h

16.08 ± 0.72l

20.09 ± 0.74m

0.00

7B1I

0,00

19,43 ± 1.01l

67,84 ± 0.58ef

62.63 ± 0.43e

21.65 ± 0.25j

67.33 ± 0.91d

0.00

0.00

40.97 ± 0.09k

7R3I

90,93 ± 1.02a

21,67 ± 0.41l

26,92 ± 1.73k

36.55 ± 0.66j

62.82 ± 0.83f

58.21 ± 0.83g

0.00

24.94 ± 0.12l

13.50 ± 0.50o

8B1I

41,70 ± 1.65f

68,96 ± 0.43e

0,73 ± 0.15pq

4.63 ± 0.28r

0.00

63.48 ± 0.41e

40.09 ± 0.36g

61.83 ± 1.36e

7.21 ± 0.26q

8R3I

0,00

13,52 ± 0.22n

0,99 ± 0.2pq

7.41 ± 0.94q

0.00

0.00

12.30 ± 2.09m

0.00

4.07 ± 0.05r

9R3I

29,15 ± 0.33g

44,41 ± 1.66i

77,24 ± 0.03b

79.38 ± 1.20b

0.00

34.81 ± 0.09k

0.00

78.31 ± 0.64a

0.00

B2Fr

47,69 ± 0.3ef

29,07 ± 0.55k

66,80 ± 1.85f

54.01 ± 0.03g

75.07 ± 1.22d

74.95 ± 0.65bc

26.04 ± 0.57i

61.99 ± 0.91e

43.70 ± 0.93j

Rb: Rhizoctonia bataticola HQ392809.1, Rs: Rhizoctonia solani KU863546, Na: Neofusicoccum australe EU375516.1, N: Nigrospora sp. JN 207298.1, B: Botryospheria sp., Fs: Fusarium solani FJ874633.1, Pu: Pythium ultimum, Fo: Fusarium oxysporum JN400698.1, C: Cylindrocarpon sp.

The phenotypic data that should be reported in table 2 are missing (see my specific comments) and in general the order in which the results are reported is confusing. The quality of the figures is low, in many cases nothing can be read. A paragraph that should be in the materials and methods is found in the results.

Corresponding author: We tried to follow reviewer recommendation as close as possible. Figures have been improved and some of them have been transferred to supplementary data. If some changes are still required, we can address them.

My suggestion is to review the article, in particular the results, reviewing the order in which they are reported, summarizing them in a clearer way and defining better the purpose of each analysis.

Corresponding author: Results part have been re-arranged. If specific reviewer changes are still required, we can revisit them.

Table 1 and many of the figures could be moved to additional materials, leaving only a few figures in the text that are fundamental to understand the results. Materials and methods can also be summarized. Many of the methods used has been already published and could be reported only as additional files to shorten the manuscript and facilitate the reading.

Corresponding author: Table 1 is transferred to supplementary material. Figures 3 B and C, Figure 6, Figure 7B have also been transferred to supplementary material.

Specific comments

Introduction:

Line 65-68: this sentence is not very clear

Corresponding author: The sentence has been rewritten.

Line 74-76: not clear

Corresponding author: Reviewer recommendation have been addressed.

Line 84: strain among species… what it means?

Corresponding author: Reviewer recommendation have been addressed.

Materials and methods

Line 100: why use 37°C for environmental strains?

Corresponding author: The temperature used in the study is the prevalent temperature in the field. Therefore, it has been selected.

Line 112: Check the names of microorganisms in many cases they are not in italic

Corresponding author: All of them are italic now.

Line 120: The number of cells used has been verified in some way?

Corresponding author: Yes, different cell numbers have been tested and the reported cell number gave the best result.

Line 172: HCN define the abbreviation

Corresponding author: It is addressed.

Line 232: remove the point, remove through filter.

Corresponding author: They are addressed.

The materials and methods for the core genome analysis are missing

Corresponding author: It is addressed.

Results:

Table 1: move to supplementary materials

Corresponding author: It is addressed.

Line 371-375: these results are not reported in table 2 as indicate by the text and I could not find these results anywhere in the manuscript

Corresponding author: Reviewer is absolutely right and data have been added to supplementary material and reported in the text.

Line 377-379: same thing the results are missing

Corresponding author: Reviewer is absolutely right and data have been added to supplementary material and reported in the text.

Figure 3B: the figure is absolutely incomprehensible, improve also the quality of figure 3C

Corresponding author: They are transferred to supplementary material.

The meaning of the two heatmaps is not clear. The numbers are the copies of genes in the genome for each category? Add a legend

Corresponding author: Reviewer is absolutely right. Therefore, we addressed the issue and added a legend.

In general, the exposure of the results is confusing. First, the authors report the phenotypic characterization of the strain, then they move to the genomic analysis, then they return to the characterization of the molecules produced, then back to the genomes. The description of the pangenoma is also confusing ... it is better to first describe the pangenoma in general and then go into details (Paragraph 3.12 should be placed before paragraph 3.11)

Corresponding author: While we understand reviewer point of view, we think that keeping the data as it is presented is better for the reader because it follows the way how the work has been conducted.

Figure 6: move to supplementary files

Corresponding author: It is addressed.

Line 454-459: materials and methods

Corresponding author: The suggested part has been removed to material and methods according to reviewer recommendation.

Figure 7B: move to supplementary files

Corresponding author: It is addressed.

Reviewer 2 Report

Cheffi et al presented an extensive manuscript about the beneficial properties of a Bacillus strain. The authors showed that this strain is able to protect olive plants from fungal and oomycetal pathogens as well as is also able to promote the olive growth and development. The authors performed the analyses both in vitro, in planta and in silico, which constitutes a proper approach and provide useful information if someone wants to use this strain in the future. Interestingly, the authors provide phylogenomics analyses in which they showed the taxonomic affiliation of this strain and defined its position among other isolates with available genomes.

However, I do have some concerns about the manuscript:

- The origin of the strain. Please, add information about the origin of the strain, place of isolation, identification, characterization….

- Figures are not where they should be within the text. You can get lost reading the manuscript. Figure and table legends should be improved as well as the quality of some of them. For example, it is impossible to read what is in figure 3 b and c. Moreover, some tables (table 1) and I think maybe some figures must be placed as supplementary data. Species column of Table 1 can be removed, as there is always the same species. Also, a column stating if the genome is complete or draft would improve the table.

-The production of CWDE is by itself considered as a PGP mechanism. There is no need to separate as endophytic traits. Moreover, the claim that this strain is an endophyte is very weak. Did the authors check if there are other strains present inside the roots (this is possible as the plants were not axenic), maybe there are strains with a natural resistance to the selected antibiotics?? Have you performed disinfection controls (i.e. surface-disinfected root passed through LB plates to see if the method was effective)? I would recommend microscopy analyses.

-Figure 1 showed a result, not a methodology.

-I would appreciate some more discussion about the taxonomic affiliation of the 4 different species that the authors suggest. It is quite interesting, and some criticism at least is expected.

- How do you extract VOCs? It is not clear, and I think not written properly in the text and them, there is a supplementary figure showing effcts of VOCs on several pathogens…

- Introduction should be improved (it is kind of short) and the methods employed should be resumed or rearranged.

Other comments:

Lines 112-120. Names must be italicized. Please, check this through the ms and also, tables and figures, as there are some errors on taxa names.

Figure 10a legends must be checked as it is difficult to follow. Panel c showed OEE1 mutant strain, only this panel? What is presented in 1, 2, 3 and 4 subpanels?

Line 537. …Mutant strain was designated with the same name as wild type?

Line 602. …rich weapons??

Please, check the species name very carefully through the ms as well as in tables and figures. Full name Bacillus velezensis should be written as B. velezensis after the first time it is named. Please, check italics as well.

Author Response

(The authors gave the same response as above.)

Academic Editor Notes

If the reviewers have suggested that your manuscript should undergo extensive English editing, please address this during revision. We suggest that you have your manuscript checked by a native English-speaking colleague or use a professional English editing service. Alternatively, MDPI provides an English editing service checking grammar, spelling, punctuation and some improvement of style where necessary for an additional charge (extensive re-writing is not included), see details at https://www.mdpi.com/authors/english.

Reviewer #2:

Comments and Suggestions for Authors

Cheffi et al presented an extensive manuscript about the beneficial properties of a Bacillus strain. The authors showed that this strain is able to protect olive plants from fungal and oomycetal pathogens as well as is also able to promote the olive growth and development. The authors performed the analyses both in vitro, in planta and in silico, which constitutes a proper approach and provide useful information if someone wants to use this strain in the future. Interestingly, the authors provide phylogenomics analyses in which they showed the taxonomic affiliation of this strain and defined its position among other isolates with available genomes.

However, I do have some concerns about the manuscript:

- The origin of the strain. Please, add information about the origin of the strain, place of isolation, identification, characterization….

Corresponding author: It is addressed.

- Figures are not where they should be within the text. You can get lost reading the manuscript. Figure and table legends should be improved as well as the quality of some of them. For example, it is impossible to read what is in figure 3 b and c. Moreover, some tables (table 1) and I think maybe some figures must be placed as supplementary data. Species column of Table 1 can be removed, as there is always the same species. Also, a column stating if the genome is complete or draft would improve the table.

Corresponding author: The comments are addressed.

-The production of CWDE is by itself considered as a PGP mechanism. There is no need to separate as endophytic traits. Moreover, the claim that this strain is an endophyte is very weak. Did the authors check if there are other strains present inside the roots (this is possible as the plants were not axenic), maybe there are strains with a natural resistance to the selected antibiotics?? Have you performed disinfection controls (i.e. surface-disinfected root passed through LB plates to see if the method was effective)? I would recommend microscopy analyses.

Corresponding author: Reviewer is absolutely right. Disinfection controls have been conducted and gave negative results. Strains with double resistance as the strain selected in the study are highly unlikely.

-Figure 1 showed a result, not a methodology.

Corresponding author: The position of Figure 1 is changed now.

-I would appreciate some more discussion about the taxonomic affiliation of the 4 different species that the authors suggest. It is quite interesting, and some criticism at least is expected.

Corresponding author: It is addressed.

- How do you extract VOCs? It is not clear, and I think not written properly in the text and them, there is a supplementary figure showing effects of VOCs on several pathogens…

Corresponding author: Corrected and explained more appropriately.

- Introduction should be improved (it is kind of short) and the methods employed should be resumed or rearranged.

Corresponding author: It is addressed.

Other comments:

Lines 112-120. Names must be italicized. Please, check this through the ms and also, tables and figures, as there are some errors on taxa names.

Corresponding author: It is addressed.

Figure 10a legends must be checked as it is difficult to follow. Panel c showed OEE1 mutant strain, only this panel? What is presented in 1, 2, 3 and 4 subpanels?

Corresponding author: Yes, reviewer is right for panel C. It is only the mutant strain.

Line 537. …Mutant strain was designated with the same name as wild type?

Corresponding author: No, it is designed by the naming OEE1 double mutant strain. In 1, 2, 3 and 4 subpanels re-isolation of the mutant strain OEE1’ from A: leaves and B: stems after root inoculation with mutant B. velezensis OEE1.

Line 602. …rich weapons??

Corresponding author: It is addressed.

Please, check the species name very carefully through the ms as well as in tables and figures. Full name Bacillus velezensis should be written as B. velezensis after the first time it is named. Please, check italics as well.

Corresponding author: It is addressed.

We would like to thank the reviewers for their valuable comments, the manuscript has now been improved. All recommendations of the reviewers have been addressed.

We would like to thank the editor for his time handing our manuscript and looking for his positive response.

Best regards

Dr. Lassaad Belbahri

Round 2

Reviewer 1 Report

The manuscript have been improved and the authors responded to all comments.

The quality of the figure have been improved. The only figure which is not yet clear is figure 4. The font of the name of the strains can not be increased but  that of functions can be increased to facilitate reading

Author Response

Reviewer 1

Open Review

English language and style

( ) Extensive editing of English language and style required 
( ) Moderate English changes required 
(x) English language and style are fine/minor spell check required 
( ) I don't feel qualified to judge about the English language and style 

Yes

Can be improved

Must be improved

Not applicable

Does the introduction provide sufficient background and include all relevant references?

(x)

( )

( )

( )

Is the research design appropriate?

(x)

( )

( )

( )

Are the methods adequately described?

(x)

( )

( )

( )

Are the results clearly presented?

(x)

( )

( )

( )

Are the conclusions supported by the results?

(x)

( )

( )

( )

Comments and Suggestions for Authors

The manuscript have been improved and the authors responded to all comments.

The quality of the figure have been improved. The only figure which is not yet clear is figure 4. The font of the name of the strains cannot be increased but that of functions can be increased to facilitate reading.

Corresponding Author: Thanks for the reviewer 1. We improved the quality of Figure 4.

Reviewer 2 Report

The manuscript is quite improved. However, there are some minor concerns that must be addressed, mostly regarding to the figures.

Figures 1, 2, 3 and 4 are out of place. Also, other figures such as figure 9 are out of place. Please, make sure that this is fixed in the last version. Figure 6 caption seems incomplete. Please, improve the caption of figure 9 a). Are there 6 strains of Fso? Or is it just one? I would recommend introducing some signal of which ones are belonging to preventive, curative and so on. There is no mention of Figure 1 in the text, just in the methods and discussion section. Figure 1 should be mentioned in section 3.1 or 3.2. Every figure must be placed after mentioned on text, in the results section, as all the figures described results and not methods. Please, fix it. Please, identify type strain(s) of B. velezensis in Figure 3, TableS2 and along the entire manuscript, when necessary. Please, clarify how the phylogenetic tree was obtained; did the authors use the core genome? The authors should indicate how many genes of the genome were considered for the tree. Why the authors did not consider all the genomes available for B. velezensis? What do the colours mean in figure 3? I suppose that every colour corresponds to a “sister clade” that represent a putative new species, but it is not written anywhere. Strain CBMB205 has two genome assemblies…is it necessary to include both in the analyses? In order to improve the interpretation and the discussion regarding to B. velezensis phylogenomic analyses, I strongly recommend the authors to check Dunlap et al 2016 (IJSEM doi: 10.1099/ijsem.0.000858).

Other comments:

There is no need of repeating the genus name “Bacillus” in the tree and in the rest of figures. Abbreviation as “B.” is best. Also, the word “strain” is not necessary to be repeated. In table 1, Toxipedia and Wikipedia references were given by the authors. I am sorry but I think this kind of references are not acceptable. Names on figure 4 are not easily readable. Could you increase letter size a bit? Line 64: “…for mean annual losses”. Do you mean “average annual losses”? Line 545: strain

Author Response

Reviewer 2

Open Review

English language and style

( ) Extensive editing of English language and style required 
( ) Moderate English changes required 
( ) English language and style are fine/minor spell check required 
(x) I don't feel qualified to judge about the English language and style 

Yes

Can be improved

Must be improved

Not applicable

Does the introduction provide sufficient background and include all relevant references?

(x)

( )

( )

( )

Is the research design appropriate?

(x)

( )

( )

( )

Are the methods adequately described?

(x)

( )

( )

( )

Are the results clearly presented?

( )

(x)

( )

( )

Are the conclusions supported by the results?

( )

(x)

( )

( )

Comments and Suggestions for Authors

The manuscript is quite improved. However, there are some minor concerns that must be addressed, mostly regarding to the figures.

Figures 1, 2, 3 and 4 are out of place. Also, other figures such as figure 9 are out of place. Please, make sure that this is fixed in the last version.

Corresponding Author: Yes Figures 1, 2, 3, 4 and 9 have been fixed according to reviewer recommendations.

Figure 6 caption seems incomplete.

Corresponding Author: Figure 6 caption is now complete following reviewer recommendation.

Please, improve the caption of figure 9 a).

Corresponding Author: Figure 9a caption is now complete following reviewer recommendation.

Are there 6 strains of Fso? Or is it just one?

Corresponding Author: There is just one (Fso1). It is addressed based on reviewer recommendation.

I would recommend introducing some signal of which ones are belonging to preventive, curative and so on.

Corresponding Author: It is improved in Figure 9a caption.

There is no mention of Figure 1 in the text, just in the methods and discussion section.

Corresponding Author: Figure 1 is now cited in the text (result section) following reviewer recommendation.

Figure 1 should be mentioned in section 3.1 or 3.2.

Corresponding Author: Figure 1 is now cited in section 3.1 and 3.2.

Every figure must be placed after mentioned on text, in the results section, as all the figures described results and not methods. Please, fix it.

Corresponding Author: Reviewer recommendation have been fulfilled. Each figure is now placed in the results section after mentioned.

Please, identify type strain(s) of B. velezensis in Figure 3, TableS2 and along the entire manuscript, when necessary.

Corresponding Author: Type strain have been identified now in Figure 3, Table S2 and along the entire manuscript following reviewer recommendation.

Please, clarify how the phylogenetic tree was obtained; did the authors use the core genome? The authors should indicate how many genes of the genome were considered for the tree.

Corresponding Author: The entire genomes parts that can align have been used for phylogenomic tree as described in the algorithm of realphy (Bertels, F.; Silander, O.K.; Pachkov, M.; Rainey, P.B.; van Nimwegen, E. Automated reconstruction of whole-genome phylogenies from short-sequence reads. Mol Biol Evol 2014, 31, 1077-1088).

Why the authors did not consider all the genomes available for B. velezensis?

Corresponding Author: At the time that the analysis has been conducted we considered all the available B. velezensis genomes. However, given the speed by which the genomes are released in the international databases, our analysis become partial soon after completion.

What do the colours mean in figure 3? I suppose that every colour corresponds to a “sister clade” that represent a putative new species, but it is not written anywhere.

Corresponding Author: Yes, reviewer is right. We indicated that in the text.

Strain CBMB205 has two genome assemblies…is it necessary to include both in the analyses?

Corresponding Author: Yes this was to check the veracity of the two assemblies and therefore validate the genome of the species.

In order to improve the interpretation and the discussion regarding to B. velezensis phylogenomic analyses, I strongly recommend the authors to check Dunlap et al 2016 (IJSEM doi: 10.1099/ijsem.0.000858).

Corresponding Author: We found this suggestion very constructive and cited this reference in the literature to meet reviewer recommendation.

Other comments:

There is no need of repeating the genus name “Bacillus” in the tree and in the rest of figures. Abbreviation as “B.” is best. Also, the word “strain” is not necessary to be repeated.

Corresponding Author: Reviewer recommendation have been fulfilled.

In table 1, Toxipedia and Wikipedia references were given by the authors. I am sorry but I think this kind of references are not acceptable.

Corresponding Author: Reviewer recommendation have been fulfilled.

Names on figure 4 are not easily readable. Could you increase letter size a bit?

Corresponding Author: Reviewer recommendation have been fulfilled.

Line 64: “…for mean annual losses”. Do you mean “average annual losses”?

Corresponding Author: Yes, reviewer is right. We replaced mean by average in the text to meet reviewer suggestion.

Line 545: strain 

Corresponding Author: Reviewer recommendation have been fulfilled.